# MABSplit: Faster Forest Training Using Multi-Armed Bandits

**Mo Tiwari**[1]  **Ryan Kang**[1,*]  **Je-Yong Lee**[2,*]  **Sebastian Thrun**[1]

**Chris Piech**[1]  **Ilan Shomorony**[#,3]  **Martin Jinye Zhang**[#,4]

## Abstract

Random forests are some of the most widely used machine learning models today, especially in domains that necessitate interpretability. We present an algorithm that accelerates the training of random forests and other popular tree-based learning methods. At the core of our algorithm is a novel node-splitting subroutine, dubbed MABSplit, used to efficiently find split points when constructing decision trees. Our algorithm borrows techniques from the multi-armed bandit literature to judiciously determine how to allocate samples and computational power across candidate split points. We provide theoretical guarantees that MABSplit improves the sample complexity of each node split from linear to logarithmic in the number of data points. In some settings, MABSplit leads to 100x faster training (an 99% reduction in training time) without any decrease in generalization performance. We demonstrate similar speedups when MABSplit is used across a variety of forest-based variants, such as Extremely Random Forests and Random Patches. We also show our algorithm can be used in both classification and regression tasks. Finally, we show that MABSplit outperforms existing methods in generalization performance and feature importance calculations under a fixed computational budget. All of our experimental results are reproducible via a one-line script at https://github.com/ThrunGroup/FastForest.

## 1 Introduction

Random Forest (RF) is a supervised learning technique that is widely used for classification and regression tasks [28, 15]. In RF, an ensemble of decision trees (DTs) is trained for the same prediction task. Each DT consists of a series of nodes that represent `if/then/else` comparisons on the feature values of a given datapoint and are used to produce an output label. In RF, each DT is typically trained on a *bootstrap* sample of the original dataset and considers a random sample of available features at each node split [14]. The prediction of the each DT is aggregated to provide an output label for the whole RF. By aggregating the prediction of each DT in the ensemble, RFs tend to be more robust to noise and overfitting [52] and are capable of capturing more complex patterns in the data than a single DT [59].

RF has gained tremendous popularity due to its flexibility, usefulness in multi-class classification and regression tasks, high performance across a broad range of data types, natural support for missing features, and relatively low computational complexity [61, 53, 10, 35, 30]. Furthermore, RF inherits

---

∗ denotes equal contribution. # denotes joint supervision. Correspondence should be addressed to M.T. (motiwari@stanford.edu), I.S. (ilans@illinois.edu), and M.J.Z. (jinyezhang@hsph.harvard.edu).

1: Department of Computer Science, Stanford University
2: Oxford University
3: Electrical and Computer Engineering, University of Illinois at Urbana-Champaign
4: Department of Epidemiology, Harvard T.H. Chan School of Public Health

36th Conference on Neural Information Processing Systems (NeurIPS 2022).

the interpretability of decision trees because the prediction of each constituent DT can be explained through a sequence of binary decisions. RFs have been successfully applied in contexts as varied as the prediction of legal court decisions [32], solar radiation analysis [10], and the Higgs Boson classification problem [3]. In the era of big data, a simple and flexible machine learning technique such as RF is expected to play a key role in processing large datasets and providing accurate and interpretable predictions.

The need for training prediction models on massive datasets and doing so on compute-constrained hardware, such as smartphones and Internet-of-Things devices, requires the development of new algorithms that can deliver faster results without sacrificing generalization performance [61]. For this reason, recent work has proposed ways to accelerate the training of RFs, both at the algorithmic level and at the hardware level.

At the algorithmic level, most work focuses on fast construction of each individual DT. Each DT is built by identifying the feature $f$ and threshold $t$ that best split the data points according to the prediction targets. The data points are split into subsets based on whether their feature $f$ has a value less than $t$ or feature $f$ has a value greater than $t$. The process is then recursed for each resulting subset. Most of the complexity in this process is in identifying the pair $(f, t)$ that provides the best split for a set of $N$ data points, which typically requires $O(N)$ computation per split. Recent proposals include computing (or estimating) $f$ and $t$ from a subsample of the data points and features, or quantizing the feature values. The latter technique creates a histogram of values of each feature across the data points and restricts $t$ to be at the edges of histogram bins.

While existing approaches provide significant speed up in the training of RFs, they often require prespecification of fixed hyperparameters, such as the proportion of data points or features to subsample, and are not adaptive to the underlying data distribution. Moreover, when comparing different candidate features for a split, all features are treated on equal footing and the quality of their split is computed based on the same number of data points. Intuitively, this is wasteful because features that are not informative for the prediction task can be identified based on a smaller number of data points. Alternatively, an adaptive scheme could better allocate computational resources towards a promising set of candidate features and achieve a better tradeoff between computational cost and generalization performance.

In this work, we propose MABSplit, a fast subroutine for the node-splitting problem, which adaptively refines the estimate of the "quality" of each feature-threshold pair $(f, t)$ as a candidate split. Bad split candidates can be discarded early, which can lead to significant computational savings. The core idea behind our algorithm is to formulate the node-splitting task as a multi-armed bandit problem [34, 2, 5, 58], where each pair $(f, t)$ is a distinct arm. The unknown parameter of each arm, $\mu_{ft}$, corresponds to the quality of the split based on feature $f$ and threshold $t$, where the split quality is measured in terms of how much the split would reduce label impurity. An arm $(f, t)$ can be "pulled" by computing the reduction of label impurity induced by a new data point sampled from the dataset. This allows us to compute an estimate $\hat{\mu}_{ft}$ and a corresponding confidence interval, which can be used in a batched variant of the Upper Confidence Bound (UCB) and successive elimination algorithms [34, 63] to identify the best arm $(f, t)$. Crucially, MABSplit uses the adaptive sampling tools of multi-armed bandits to avoid computing the split qualities over the entire dataset.

We demonstrate the benefits of MABSplit on a variety of datasets, for both classification and regression tasks. In some settings, MABSplit algorithm leads to 100x faster training (a 99% reduction in training time), without any decrease in test accuracy, over an exact implementation of RF that searches for the optimal $(f, t)$ pair via brute-force computation. Additionally, we demonstrate similar speedups when using MABSplit across a variety of forest-based variants, such as Extremely Random Forests and Random Patches.

## 1.1 Related work

Random Forests were originally proposed by Ho [28]. In its original formulation, RF constructs $n_{\text{tree}}$ DTs, where each DT is trained on a bootstrap sample of all $N$ data points and a random subset of the features at each node (a technique known as random subspacing [29]). More recently, the need for training RFs on large datasets has prompted the development of several techniques to accelerate training at both the software and hardware levels.

**Software acceleration of RF:** Most of the software and algorithmic acceleration techniques focus on the training of each individual DT. FastForest [61] accelerates the node-splitting task using three ideas: subsampling a pre-specified number of data points without replacement (subbagging), subsampling a pre-specified number of features dependent on the current number of data points (Dynamic Restricted Subspacing), and dividing values of a given feature into $T$ bins, where $T$ depends on the number of data points at the node (Logarithmic Split-Point Sampling, inspired by the single-tree SPAARC algorithm [62]). MABSplit is inspired by the ideas in FastForest, but does not require the number of data points or features to be prespecified and, instead, determines them by adapting to the data distribution.

Other recent work has also used adaptivity to identify the best split. For example, Very Fast Decision Trees (VFDTs) [19] and Extremely Fast Decision Trees (EFDTs) [40] are incremental decision tree learning algorithms in which trees can be updated in streaming settings. Acceleration of the node-splitting task is achieved by adaptively selecting a subset of data points sufficient to distinguish the best and second best splits. These approaches are similar to ours, but the sampled data points are used to evaluate all possible splits. MABSplit, in contrast, adaptively discards unpromising splits early. The F-forest [23] algorithm also applies adaptivity and uses an upper bound on the impurity reduction of each split in order to discard candidate splits. This is similar in spirit to the goal of MABSplit, but is based on a deterministic, conservative upper bound on the impurity reduction (as opposed to MABSplit's statistical estimate) and incurs computation linear in the node size for each split, even when considering a fixed number of possible split thresholds per feature.

Other variations of RF have been proposed to improve training time. Random Patches [39] builds trees based on a subset of data points and features that is fixed for each entire forest. ExtraTrees (ETs) [24] draw a random subset of $K$ features at each node and, for each one, chooses a number $R$ of random splits. It then selects the split that yields the largest impurity reduction from among these $KR$ candidate splits.

Other recent work attempts to accelerate RF training by identifying the optimal number of decision trees needed in the forest [45], a form of hyperparameter tuning. The MABSplit subroutine can also be incorporated into these methods.

**Hardware acceleration of RF:** The training of RFs can also be significantly accelerated through the use of specialized hardware. For instance, the implementation of RF available in Weka [21] allows trees to be trained on different cores and reduces forest training time. A GPU-based parallel implementation of RF has also been proposed in [26]. These solutions require specialized hardware (e.g., GPU-based PC video cards) and are inappropriate for everyday users locally executing data-mining tasks on standard PCs or smartphones. As such, it is still desirable to develop techniques to improve prediction performance and processing speed at an algorithmic, platform-independent level.

## 2    Preliminaries

We now formally describe the RF algorithm and other tree-based models, all of which rely on a node-splitting subroutine. We consider $N$ data points $\{(\mathbf{x}_i, y_i)\}_{i=1}^N$ where each $\mathbf{x}_i$ is an $M$-dimensional feature vector and $y_i$ is its target. Following standard literature, we consider flexible feature types such as numerical or categorical. We consider both categorical targets for classification and numerical targets for regression. With a slight abuse of notation, we use $\mathcal{X}$ to mean either the set of indices $\{i\}$ or the values $\{(\mathbf{x}_i, y_i)\}$, with the meaning clear from context.

An RF contains $n_{\text{tree}}$ decision trees, each trained on a set of $N$ bootstrapped data points (sampled with replacement) and a random subset of features at each node. The whole RF, an ensemble model, outputs the trees' majority vote in classification and the trees' average prediction in regression [28, 15]. We focus on the top-down approach of constructing DTs by choosing the feature-threshold pair at each step that best splits the set of targets at a given node [52].

We now discuss the method by which the best split is chosen in each node. Consider a node with $n$ data points $\mathcal{X}$ and $m$ features $\mathcal{F}$. Note that $n$ and $m$ at the given node may or may not be the same as $N$ and $M$ of the entire dataset (for example, RF considers a random subset of $m = \sqrt{M}$ features at each node [12]). Let $\mathcal{X}_{\text{L}, ft}$ and $\mathcal{X}_{\text{R}, ft}$ be the left and right child subsets of $\mathcal{X}$ when $\mathcal{X}$ is split according to feature $f$ at threshold $t$. The approach finds the split that best reduces the label

impurity; i.e., finds

$$f^*, t^* = \underset{f \in \mathcal{F}, t \in \mathcal{T}_f}{\arg\min} \frac{|\mathcal{X}_{\text{L},ft}|}{n} I(\mathcal{X}_{\text{L},ft}) + \frac{|\mathcal{X}_{\text{R},ft}|}{n} I(\mathcal{X}_{\text{R},ft}) - I(\mathcal{X}), \tag{1}$$

where $I(\mathcal{S})$ measures the impurity of targets $\{y_i\}_{i \in \mathcal{S}}$ and $\mathcal{T}_f$ is the set of allowed thresholds for feature $f$. Common impurity measures include Gini impurity or entropy for classification, and mean-squared-error (MSE) for regression [16]:

$$\text{Gini} : 1 - \sum_{k=1}^{K} p_k^2, \quad \text{Entropy} : - \sum_{k=1}^{K} p_k \log_2 p_k, \quad \text{MSE} : \frac{1}{n} \sum_{i \in \mathcal{X}} (y_i - \bar{y})^2, \tag{2}$$

where $K$ is the total number of classes and $p_k = \frac{1}{n} \sum_{i \in \mathcal{X}} \mathbb{I}(y_i = k)$ is the proportion of class $k$ in $\mathcal{X}$ in classification, and $\bar{y} = \frac{1}{n} \sum_{i \in \mathcal{X}} y_i$ is the average target value in regression. We note that our proposed algorithm, MABSplit, does not assume any particular structure of $I(\cdot)$.

While the conventional RF considers all $(n-1)$ possible splits among the $n$ generally different values in the dataset for a given feature $f$, in this work we focus on the histogram-based variant that chooses the threshold from a set of predefined values $\mathcal{T}_f$, e.g., $|\mathcal{T}_f|$ equally-spaced histogram bin edges; this variant is substantially more efficient, offers comparable accuracy, and has been used in most state-of-the-art implementations such as XGBoost and LightGBM [51, 17, 33, 62]. A naïve algorithm finds the best feature-threshold pairs in Equationh (1) by evaluating the label impurity reduction for each feature-threshold over all $n$ data points, which incurs computation linear in $n$.

## 3 MABSplit

We now discuss MABSplit and how it can reduce the complexity of the node-splitting problem to logarithmic in $n$.

With the same notation as in Section 2, we note that Equation (1) is equivalent to

$$f^*, t^* = \underset{f \in \mathcal{F}, t \in \mathcal{T}_f}{\arg\min} \frac{|\mathcal{X}_{\text{L},ft}|}{n} I(\mathcal{X}_{\text{L},ft}) + \frac{|\mathcal{X}_{\text{R},ft}|}{n} I(\mathcal{X}_{\text{R},ft}) \tag{3}$$

and so we focus on solving Equation (3). Let $\mu_{ft} = \frac{|\mathcal{X}_{\text{L},ft}|}{n} I(\mathcal{X}_{\text{L},ft}) + \frac{|\mathcal{X}_{\text{R},ft}|}{n} I(\mathcal{X}_{\text{R},ft})$ be the optimization objective for feature-threshold pair $(f, t)$. Omitting the dependence on $K$, computing $\mu_{ft}$ exactly is at least $O(n)$. MABSplit, however, *estimates* $\mu_{ft}$ with less computation by drawing $n' < n$ independent samples with replacement from $\mathcal{X}$. As shown in Subsec. 3.1, it is possible to construct a point estimate $\hat{\mu}_{ft}(n')$ and a $(1 - \delta)$ confidence interval (CI) $C_{ft}(n', \delta)$ for the parameter $\mu_{ft}$, where $n'$ and $\delta$ determine estimation accuracy. The width of these CIs generally scales with $\sqrt{\frac{\log 1/\delta}{n'}}$. To estimate the solution to Problem (3) with high confidence, we can then choose to sample different amounts of data points for different features so as to estimate their impurity reductions to varying degrees of accuracy. Intuitively, promising features-threshold splits with high impurity reductions (lower values of $\mu_{ft}$) should be estimated with high accuracy with many data points, while less promising ones with low impurity reductions (higher values of $\mu_{ft}$) can be discarded early.

The exact adaptive estimation procedure, MABSplit, is described in Algorithm 1. It can be viewed as a batched version of the conventional UCB algorithm [34, 63] combined with successive elimination [22], is straightforward to implement, and has been used in other applications [58, 55, 56, 6, 7]. Algorithm 1 uses the set $\mathcal{S}_{\text{solution}}$ to track all potential solutions to Problem (3); $\mathcal{S}_{\text{solution}}$ is initialized as the set of all feature-threshold pairs $\{(f, t)\}$ and Algorithm 1 maintains the mean objective estimate $\hat{\mu}_{ft}$ and $(1 - \delta)$ CI $C_{ft}$ for each potential solution $(f, t) \in \mathcal{S}_{\text{solution}}$.

In each iteration, a new batch of data points $\mathcal{X}_{\text{batch}}$ is used to evaluate the split quality for all potential feature-threshold splits in $\mathcal{S}_{\text{solution}}$, which allows the estimate of each $\hat{\mu}_{ft}$ to be made more accurate. Based on the current estimate, if a candidate's lower confidence bound $\hat{\mu}_{ft} - C_{ft}$ is greater than the upper confidence bound of the most promising candidate $\min_{f,t}(\hat{\mu}_{ft} + C_{ft})$, we remove it from $\mathcal{S}_{\text{solution}}$. This process continues until there is only one candidate in $\mathcal{S}_{\text{solution}}$ or until we have sampled more than $n$ data points. In the latter case, we know that the difference between the remaining candidates in $\mathcal{S}_{\text{solution}}$ is so subtle that an exact computation is warranted. MABSplit then compute those candidates' objectives exactly and returns the best candidate in the set.

## 3.1 Point estimates and confidence intervals for impurity metrics

We now discuss MABSplit's construction of point estimates and confidence intervals of $\mu_{ft}$ based on a set of $n'$ points, $\{(\mathbf{X}_i, Y_i)\}_{i=1}^{n'}$, sampled independently and with replacement from $\mathcal{X}$. We consider two widely used impurity metrics in classification, Gini impurity and entropy, although mean estimates and confidence intervals for other settings and metrics can be derived similarly (more details are provided in Appendix 3).

Let $p_{\mathrm{L},k} := \frac{1}{n}\sum_{i=1}^{n} \mathbb{I}(x_{if} < t, y_i = k)$ and $p_{\mathrm{R},k} := \frac{1}{n}\sum_{i=1}^{n} \mathbb{I}(x_{if} \geq t, y_i = k)$ denote the proportion of the full $n$ data points in class $k$ and each of the two subsets created by the split $(f, t)$ (we call these subsets "left" and "right", respectively). Note that

$$\sum_k p_{\mathrm{L},k} = \frac{|\mathcal{X}_{\mathrm{L},ft}|}{n} \quad \text{and} \quad \sum_k p_{\mathrm{R},k} = \frac{|\mathcal{X}_{\mathrm{R},ft}|}{n}. \tag{4}$$

Furthermore, let $\hat{p}_{\mathrm{L},k} := \frac{1}{n'}\sum_{i=1}^{n'} \mathbb{I}(X_{if} < t, Y_i = k)$ and $\hat{p}_{\mathrm{R},k} := \frac{1}{n'}\sum_{i=1}^{n'} \mathbb{I}(X_{if} \geq t, Y_i = k)$ denote the empirical estimates of $p_{\mathrm{L},k}$ and $p_{\mathrm{R},k}$ based on the $n'$ subsamples drawn thus far. Then $\{\hat{p}_{\mathrm{L},k}, \hat{p}_{\mathrm{R},k}\}_{k=1}^{K}$ jointly follow a multinomial distribution satisfying

$$\mathbb{E}[\hat{p}_{\mathrm{L},k}] = p_{\mathrm{L},k}, \quad \mathrm{Var}[\hat{p}_{\mathrm{L},k}] = \frac{1}{n'} p_{\mathrm{L},k}(1 - p_{\mathrm{L},k}),$$

$$\mathbb{E}[\hat{p}_{\mathrm{R},k}] = p_{\mathrm{R},k}, \quad \mathrm{Var}[\hat{p}_{\mathrm{R},k}] = \frac{1}{n'} p_{\mathrm{R},k}(1 - p_{\mathrm{R},k}).$$

since for each random data point $(\mathbf{X}_i, Y_i)$, exactly one element of the set $\{\hat{p}_{\mathrm{L},k}, \hat{p}_{\mathrm{R},k}\}_{k=1}^{K}$ is incremented. Using Equations (2) and (4), and the definition of $\mu_{ft}$ after Equation (3), we write

$$\text{Gini impurity}: \ \mu_{ft} = 1 - \frac{\sum_k p_{\mathrm{L},k}^2}{\sum_k p_{\mathrm{L},k}} - \frac{\sum_k p_{\mathrm{R},k}^2}{\sum_k p_{\mathrm{R},k}}, \tag{5}$$

$$\text{Entropy}: \ \mu_{ft} = -\sum_k p_{\mathrm{L},k} \log_2 \frac{p_{\mathrm{L},k}}{\sum_{k'} p_{\mathrm{L},k'}} - \sum_k p_{\mathrm{R},k} \log_2 \frac{p_{\mathrm{R},k}}{\sum_{k'} p_{\mathrm{R},k'}}, \tag{6}$$

where we can use the empirical parameters $\{\hat{p}_{\mathrm{L},k}, \hat{p}_{\mathrm{R},k}\}_{k=1}^{K}$ as plug-in estimators for the true parameters $\{p_{\mathrm{L},k}, p_{\mathrm{R},k}\}_{k=1}^{K}$ to produce the point estimate $\hat{\mu}_{ft}$.

In MABSplit (Algorithm 1), each batch of $B$ data points is used to update each $\hat{p}_{\mathrm{L},k}$ and $\hat{p}_{\mathrm{R},k}$, which are used in turn to update our point estimates $\hat{\mu}_{ft}$ and the corresponding CIs. The CIs of $\hat{\mu}_{ft}$ are based on standard error derived using the delta method [60]. As in standard applications of the delta method, the estimates $\hat{\mu}_{ft}$ are asymptotically unbiased and their corresponding CIs are asymptotically valid. Appendix 3 provides further details, including a derivation of the CIs and discussion of convergence properties.

## 3.2 Algorithmic and implementation details

We considered sampling with replacement in MABSplit (Algorithm 1) primarily for the ease of theoretical analysis. In practice, we found sampling without replacement was more computationally efficient and did not significantly change the results, and was used in the actual implementation.

For DTs, in classification tasks, we allow individual DTs to provide soft votes and average their predicted class probabilities to determine the forest's predicted class label, following existing approaches [47]. This is in contrast with hard votes, in which each DT is only permitted to produce its best label and the forest's prediction is determined by majority voting. For the fixed-budget experiments in Section 5, we terminate tree construction and do not split nodes further if doing so would violate our budget constraints.

## 4 Analysis of the Algorithm

In this section, we prove that MABSplit returns the optimal feature-threshold pair for a node split with high probability. Furthermore, we provide bounds on computational complexity of MABSplit, which can lead to a logarithmic dependence on the number of data points $n$ under weak assumptions.

---

**Algorithm 1** MABSplit ( $\mathcal{X}, \mathcal{F}, \mathcal{T}_f, I(\cdot), B, \delta$ )

---
1:  $\mathcal{S}_{\text{solution}} \leftarrow \{(f,t), \forall f \in \mathcal{F}, \forall t \in \mathcal{T}_f\}$      ▷ Set of potential solutions to Problem (3)
2:  $n_{\text{used}} \leftarrow 0$      ▷ Number of data points sampled
3:  For all $(f,t) \in \mathcal{S}_{\text{solution}}$, set $\hat{\mu}_{ft} \leftarrow \infty, C_{ft} \leftarrow \infty$      ▷ Initialize mean and CI for each arm
4:  **for all** $f \in \mathcal{F}$ **do**
5:       Create empty histogram, $h_f$, with $|\mathcal{T}_f| = T$ equally spaced bins
6:  **while** $n_{\text{used}} < n$ and $|\mathcal{S}_{\text{solution}}| > 1$ **do**
7:       Draw a batch sample $\mathcal{X}_{\text{batch}}$ of size $B$ with replacement from $\mathcal{X}$
8:       **for all** unique $f$ in $\mathcal{S}_{\text{solution}}$ **do**
9:           **for all** $x$ in $\mathcal{X}_{\text{batch}}$ **do**
10:              Insert $x_f$ into histogram $h_f$      ▷ Each insertion is $O(1)$
11:      **for all** $(f,t) \in \mathcal{S}_{\text{solution}}$ **do**
12:          Update $\hat{\mu}_{ft}$ and $C_{ft}$ based on histogram $h_f$      ▷ For fixed $f$, this is $O(T)$
13:      $\mathcal{S}_{\text{solution}} \leftarrow \{(f,t) : \hat{\mu}_{ft} - C_{ft} \leq \min_{f,t}(\hat{\mu}_{ft} + C_{ft})\}$      ▷ Retain only promising splits
14:      $n_{\text{used}} \leftarrow n_{\text{used}} + B$
15: **if** $|\mathcal{S}_{\text{solution}}| = 1$ **then**
16:      **return** $(f^*, t^*) \in \mathcal{S}_{\text{solution}}$
17: **else**
18:      Compute $\mu_{ft}$ exactly for all $(f,t) \in \mathcal{S}_{\text{solution}}$
19:      **return** $(f^*, t^*) = \arg\min_{(f,t) \in \mathcal{S}_{\text{solution}}} \mu_{ft}$

---

As above, consider a node with $n$ data points $\mathcal{X}$, $m$ features $\mathcal{F}$, and $T$ possible thresholds for each feature ($|\mathcal{T}_f| = T$ for all $f$). Suppose $(f^*, t^*) = \arg\min_{f \in \mathcal{F}, t \in \mathcal{T}_f} \mu_{ft}$ is the optimal feature-threshold pair at which to split the node. For any other feature-threshold pair $(f,t)$, define $\Delta_{ft} :=  \mu_{ft} - \mu_{f^*t^*}$. To state the following results, we will assume that, for a fixed feature-threshold pair $(f,t)$ and $n'$ randomly sampled datapoints, the $(1 - \delta)$ confidence interval scales as $C_{ft}(n', \delta) = O(\sqrt{\frac{\log 1/\delta}{n'}})$. (This assumption is justified for the confidence intervals of Gini impurity and entropy under weak assumptions on the $\mu_{f,t}$'s [60]). With this assumption, we state the following theorem:

**Theorem 1.** *Assume $\exists c_0 > 0$ s.t. $\forall \delta > 0, n' > 0$, we have $C_{ft}(n', \delta) < c_0 \sqrt{\frac{\log 1/\delta}{n'}}$. For $\delta = \frac{1}{n^2 mT}$, with probability at least $1 - \frac{1}{n}$, Algorithm 1 returns the correct solution to Equation (3). Furthermore, Algorithm 1 uses a total of $M$ computations, where*

$$\mathbb{E}[M] \leq \sum_{f \in \mathcal{F}, t \in \mathcal{T}_f} \min\left[\frac{4c_0^2}{\Delta_{ft}^2} \log(n^2 mT) + B, 2n\right] + 2mT. \tag{7}$$

Theorem 1 is proven in Appendix 1. Intuitively, Theorem 1 states that with high probability, MABSplit returns the optimal feature-threshold pair at which to split the node. The bound Equation (7) suggests the computational cost of a feature-threshold pair $(f,t)$, i.e., $\min\left[\frac{4c_0^2}{\Delta_{ft}^2} \log(n^2 mT) + B, 2n\right]$, depends on $\Delta_{ft}$, which measures how close its optimization parameter $\mu_{ft}$ is to $\mu_{f^*t^*}$. Most reasonably different features $f \neq f^*$ will have a large $\Delta_{ft}$ and incur computational cost $O(\log(n^2 mT))$ that is sublinear in $n$.

In turn, this implies MABSplit takes only $O(mT \log(n^2 mT))$ computations per feature-threshold pair if there is reasonable heterogeneity among them. As proven in Appendix 2 of [5], this is the case under a wide range of distributional assumptions on the $\mu_{ft}$'s, e.g., when the $\mu_{ft}$'s follow a sub-Gaussian distribution across the pairs $(f,t)$. Such assumptions ensure that MABSplit has an overall complexity of $O(mT \log(n^2 mT))$, which is sublinear in the number of data points $n$. We note that in the worst case, however, MABSplit may take $O(n(m + T))$ computations per feature-threshold pair when most splits are equally good, in which case MABSplit reduces to a batched version of the naïve algorithm. This may happen, for example, in highly symmetric datasets where all splits reduce the impurity equally. Other recent work provides further discussion on the conversion between a bound like Equation (7), which depends on the $\Delta_i$'s, and a bound in terms of other problem parameters such as $O(mT \log(n^2 mT))$ under various assumptions on the $\mu_{ft}$'s [5, 63, 7, 58, 4, 6].

Finally, we note that $\delta$ is a hyperparameter governing the error rate. It is possible to prove results analogous to Theorem 1 for arbitrary $\delta$.

## 5   Experimental Results

We demonstrate the advantages of MABSplit in two settings. In the first setting, the baseline models with and without MABSplit are trained to completion and we report wall-clock training time and generalization performance. In the second setting, we consider training each forest with a fixed computational budget and study effect of MABSplit on generalization performance. We provide a description of each dataset in Appendix 6.

We note that head-to-head wall-clock time comparisons with common implementations of forest-based algorithms, such as Weka's [21] or `scikit-learn`'s [48], would be unfair due to their extensive hardware- and language-level optimizations. As such, we reimplement these baselines in Python and focus on algorithmic improvements. The only difference between our model and the baselines is the call to the node-splitting subroutine (MABSplit in our model, and the exact, brute-force solver for the baseline models); thus, any improvements in runtime are due to improvements in the node-splitting algorithm. This is verified by profiling the implementations and measuring the relative time spent in the node-splitting subroutine versus total runtime (see Appendix 5). Our approach allows us to focus on algorithmic improvements as opposed implementation-specific optimizations. Our implementation of these baselines may also be of independent interest and we verify the quality of our implementations, via agreement with `scikit-learn`, in Appendix 4. We also provide a brief discussion of how an optimized version of our reimplementations may outperform `scikit-learn` in Appendix 7. On the MNIST dataset, our optimized implementation trains approximately 4x faster than `scikit-learn`'s `DecisionTreeClassifier`.

**Baseline Models:** We compare the histogrammed versions of three baselines with and without MABSplit: Random Forest (RF), ExtraTrees [24], and Random Patches (RP) [39]. In Random Forests, each tree fits a bootstrap sample of $N$ datapoints and considers random subset of $\sqrt{M}$ features at every node split. Extra Trees (also known as Extremely Randomized Forests) are identical to Random Forests except for two differences. First, in regression problems, all features are considered at every node split (in classification problems, we still use only $\sqrt{M}$ features). Second, the histogram edges of a feature are randomly chosen from a uniform distribution over that feature's minimum and maximum value. In classification problems, each histogram has $\sqrt{M}$ bins and in regression problems, each histogram has $M$ bins. These conventions follow standard implementations [47]. Note that in ExtraTrees, the bins in a feature's histogram need not be equally spaced. Random Patches is identical to Random Forests but the training dataset is reduced to $\alpha_n$ of its original datapoints and $\alpha_f$ of its original features, where $\alpha_n$ and $\alpha_f$ are prespecified constants, and the subsampled dataset is fixed for the training of the entire forest. Full settings for all experiments are given in Appendix 6.

### 5.1   Wall-clock time comparisons

In the first setting, we compare baseline models with and without MABSplit in terms of wall-clock training time. Tables 1 and 2 show that MABSplit provides similar generalization performance but faster training than the usual naïve algorithm for node-splitting for almost all baselines in both classification and regression tasks. Across various tasks, MABSplit leads to approximately 2x-100x faster training, a reduction of training time of 50-99%. These benefits are wholly attributable to MABSplit as the only difference between successive minor rows in each table is the node-splitting subroutine (see Appendix 5 for further discussion).

### 5.2   Fixed budget comparisons

In the second setting, we consider training models under a fixed computational budget. As before, insertion into a histogram is taken to be an $O(1)$ operation. This is justified if the histogram's thresholds are evenly spaced, wherefore the correct bin in which to insert a value can be indexed into directly. (If the bins are unevenly spaced, we may perform binary searches to locate the correct bin, which is $O(\log T)$ and does not depend on $n$, or cache the results of these binary searches for an evenly-spaced grid across the range of the given feature's value.)

| MNIST Dataset ($N = 60,000$) | | | |
|---|---|---|---|
| Model | Training Time (s) | Number of Insertions | Test Accuracy |
| RF | 1542.83 ± 5.837 | 1.44E+08 ± 4.85E+05 | 0.777 ± 0.005 |
| **RF + MABSplit** | **40.359 ± 0.246** | **3.37E+06 ± 1.62E+04** | **0.763 ± 0.008** |
| ExtraTrees | 1789.653 ± 2.396 | 1.68E+08 ± 0.00E+00 | 0.762 ± 0.003 |
| **ExtraTrees + MABSplit** | **50.217 ± 0.304** | **4.32E+06 ± 7.69E+03** | **0.755 ± 0.002** |
| RP | 1421.963 ± 8.368 | 1.32E+08 ± 6.95E+05 | 0.771 ± 0.003 |
| **RP + MABSplit** | **38.415 ± 0.245** | **3.17E+06 ± 1.40E+04** | **0.768 ± 0.003** |
| APS Failure at Scania Trucks Dataset ($N = 60,000$) | | | |
| Model | Training Time (s) | Number of Insertions | Test Accuracy |
| RF | 20.542 ± 0.048 | 3.77E+06 ± 9.66E+03 | 0.985 ± 0.0 |
| **RF + MABSplit** | **0.455 ± 0.002** | **6.94E+04 ± 2.19E+02** | **0.985 ± 0.0** |
| ExtraTrees | 18.849 ± 0.027 | 3.78E+06 ± 0.00E+00 | 0.985 ± 0.0 |
| **ExtraTrees + MABSplit** | **0.406 ± 0.001** | **7.00E+04 ± 0.00E+00** | **0.985 ± 0.0** |
| RP | 17.63 ± 0.054 | 3.22E+06 ± 1.18E+04 | 0.985 ± 0.0 |
| **RP + MABSplit** | **0.399 ± 0.003** | **5.96E+04 ± 2.19E+02** | **0.985 ± 0.0** |
| Forest Covertype Dataset ($N = 581,012$) | | | |
| Model | Training Time (s) | Number of Insertions | Test Accuracy |
| RF | 117.351 ± 0.123 | 1.86E+07 ± 0.00E+00 | 0.559 ± 0.028 |
| **RF + MABSplit** | **0.88 ± 0.009** | **3.98E+04 ± 1.79E+02** | **0.505 ± 0.004** |
| ExtraTrees | 117.984 ± 0.119 | 1.86E+07 ± 0.00E+00 | 0.539 ± 0.022 |
| **ExtraTrees + MABSplit** | **2.942 ± 0.856** | **3.69E+05 ± 1.22E+05** | **0.5 ± 0.005** |
| RP | 104.456 ± 0.737 | 1.62E+07 ± 8.31E+04 | 0.51 ± 0.008 |
| **RP + MABSplit** | **0.815 ± 0.004** | **3.50E+04 ± 0.00E+00** | **0.507 ± 0.005** |

Table 1: Wall-clock training time, number of histogram insertions, and test accuracies for various models with and without MABSplit. MABSplit can accelerate these models by over 100x in some cases (an 99% reduction in training time) while achieving comparable accuracy. The number of histogram insertions correlates strongly with wall-clock training time, which justifies our focus on accelerating the node-splitting algorithm via reductions in sample complexity.

| Beijing Multi-Site Air-Quality Dataset (Regression, $N = 420,768$) | | |
|---|---|---|
| Model | Training Time (s) | Test MSE |
| RF | 138.782 ± 1.581 | 1164.576 ± 0.761 |
| **RF + MABSplit** | **67.089 ± 1.682** | **1109.542 ± 23.776** |
| ExtraTrees | 115.592 ± 3.061 | 1028.054 ± 11.355 |
| **ExtraTrees + MABSplit** | **53.607 ± 1.278** | **1015.234 ± 6.535** |
| RP | 108.174 ± 1.24 | 1128.299 ± 25.78 |
| **RP + MABSplit** | **60.639 ± 3.642** | **1125.816 ± 33.56** |
| SGEMM GPU Kernel Performance Dataset (Regression, $N = 241,600$) | | |
| Model | Training Time (s) | Test MSE |
| RF | 32.606 ± 0.859 | 69733.002 ± 57.401 |
| **RF + MABSplit** | **16.51 ± 0.224** | **69493.921 ± 73.133** |
| ExtraTrees | 30.624 ± 0.686 | 69734.948 ± 54.876 |
| **ExtraTrees + MABSplit** | **14.086 ± 0.295** | **69585.029 ± 80.281** |
| RP | 26.091 ± 0.417 | 66364.998 ± 894.568 |
| **RP + MABSplit** | **16.409 ± 0.952** | **66310.138 ± 896.237** |

Table 2: Wall-clock training time and test MSEs for various models with and without MABSplit. MABSplit can accelerate these models by up to 2x (an 50% reduction in training time) while achieving comparable results. We omit the number of histogram insertions in favor of wall-clock time for simplicity; unlike in classification, the different baseline regression models have widely varying histogram bin counts. Since the histogram insertion complexity is different across models, the comparison across models would not be fair.

| MNIST Dataset ($N = 60,000$) | | |
|---|---|---|
| Model | Number of Trees | Test Accuracy |
| RF | $0.2 \pm 0.179$ | $0.143 \pm 0.026$ |
| **RF + MABSplit** | **$15.8 \pm 0.179$** | **$0.83 \pm 0.002$** |
| ExtraTrees | $0.2 \pm 0.179$ | $0.144 \pm 0.027$ |
| **ExtraTrees + MABSplit** | **$12.0 \pm 0.0$** | **$0.814 \pm 0.001$** |
| RP | $1.0 \pm 0.0$ | $0.253 \pm 0.003$ |
| **RP + MABSplit** | **$16.8 \pm 0.179$** | **$0.832 \pm 0.002$** |
| APS Failure at Scania Trucks Dataset ($N = 60,000$) | | |
| Model | Number of Trees | Test Accuracy |
| RF | $1.0 \pm 0.0$ | $0.985 \pm 0.0$ |
| **RF + MABSplit** | **$5.8 \pm 0.179$** | **$0.989 \pm 0.0$** |
| ExtraTrees | $1.0 \pm 0.0$ | $0.985 \pm 0.0$ |
| **ExtraTrees + MABSplit** | **$5.6 \pm 0.219$** | **$0.989 \pm 0.0$** |
| RP | $1.0 \pm 0.0$ | $0.985 \pm 0.0$ |
| **RP + MABSplit** | **$6.8 \pm 0.179$** | **$0.989 \pm 0.0$** |
| Forest Covertype Dataset ($N = 581,012$) | | |
| Model | Number of Trees | Test Accuracy |
| RF | $0.4 \pm 0.219$ | $0.514 \pm 0.019$ |
| **RF + MABSplit** | **$99.8 \pm 0.179$** | **$0.675 \pm 0.002$** |
| ExtraTrees | $0.2 \pm 0.179$ | $0.496 \pm 0.006$ |
| **ExtraTrees + MABSplit** | **$23.4 \pm 1.403$** | **$0.677 \pm 0.002$** |
| RP | $0.6 \pm 0.219$ | $0.534 \pm 0.03$ |
| **RP + MABSplit** | **$100.0 \pm 0.0$** | **$0.675 \pm 0.002$** |

Table 3: Classification performance under a fixed computational budget (number of histogram insertions) for various models with and without MABSplit. MABSplit allows for more trees to be trained and leads to better generalization performance.

Intuitively, the MABSplit algorithm allows for splitting a given node with less data point queries and histogram insertions than the naïve solution. As such, when the computational budget is fixed, forests trained with MABSplit should be able to split more nodes and therefore train more trees than forests trained with the naïve solver. Prior work suggests that increasing the number of trees in a forest improves generalization performance by reducing variance at the cost of slightly increased bias [27].

Tables 3 and 4 demonstrate the generalization performance of different models as the computational budget is held constant for different classification and regression tasks. When using MABSplit, the trained forests consist of more trees and demonstrate better generalization performance across all baseline models.

### 5.3 Feature stability comparisons

We also apply MABSplit to compute feature importances under a fixed budget. We follow the common approach of computing the feature importances of multiple forests and then measuring the stability of feature selection across forests using Permutation Feature Importance and Mean Decrease in Impurity (MDI) [42, 49] (see Appendix 6 for a further discussion of these metrics). The forests trained with MABSplit demonstrate better feature stabilities than those trained with the naïve algorithm; see Table 5. Note that the datasets used for these experiments are different from the real-world datasets used in the other experiments and are described in Appendix 6.

## 6 Discussions and Conclusions

In this work, we presented a novel algorithm, MABSplit, for determining the optimal feature and corresponding threshold at which to split a node in tree-based learning models. Unlike prior models such as Random Patches, in which the subsampling hyperparameters $\alpha_n$ and $\alpha_f$ must be prespecified manually, MABSplit requires no tuning and queries only as much data as is needed by virtue of its

| Beijing Multi-Site Air-Quality Dataset ($N = 420,768$) | | |
|---|---|---|
| Model | Number of Trees | Test MSE |
| RF | $0.0 \pm 0.0$ | $3208.93 \pm 0.0$ |
| **RF + MABSplit** | **$12.0 \pm 0.0$** | **$927.013 \pm 2.042$** |
| RP | $0.0 \pm 0.0$ | $3208.93 \pm 0.0$ |
| **RP + MABSplit** | **$11.0 \pm 0.4$** | **$875.764 \pm 3.064$** |
| ExtraTrees | $0.0 \pm 0.0$ | $3208.93 \pm 0.0$ |
| **ExtraTrees + MABSplit** | **$9.0 \pm 0.0$** | **$834.338 \pm 4.377$** |
| SGEMM GPU Kernel Performance Dataset ($N = 241,600$) | | |
| Model | Number of Trees | Test MSE |
| RF | $0.0 \pm 0.0$ | $131323.839 \pm 0.0$ |
| **RF + MABSplit** | **$5.6 \pm 0.219$** | **$28571.393 \pm 357.433$** |
| RP | $0.8 \pm 0.179$ | $102616.047 \pm 6647.02$ |
| **RP + MABSplit** | **$2.8 \pm 0.593$** | **$64876.329 \pm 13350.921$** |
| ExtraTrees | $0.0 \pm 0.0$ | $131323.839 \pm 0.0$ |
| **ExtraTrees + MABSplit** | **$5.0 \pm 0.0$** | **$29919.254 \pm 344.409$** |

Table 4: Regression performance under a fixed computational budget (number of histogram insertions). MABSplit allows for more trees to be trained and leads to better generalization performance.

| Importance Model | Stability Metric | Dataset | Stability |
|---|---|---|---|
| RF | MDI | Random Classification | $0.536 \pm 0.039$ |
| **RF + MABSplit** | **MDI** | **Random Classification** | **$0.863 \pm 0.016$** |
| RF | MDI | Random Regression | $0.134 \pm 0.021$ |
| **RF + MABSplit** | **MDI** | **Random Regression** | **$0.674 \pm 0.043$** |
| RF | Permutation | Random Classification | $0.579 \pm 0.023$ |
| **RF + MABSplit** | **Permutation** | **Random Classification** | **$0.69 \pm 0.023$** |
| RF | Permutation | Random Regression | $0.116 \pm 0.017$ |
| **RF + MABSplit** | **Permutation** | **Random Regression** | **$0.437 \pm 0.044$** |

Table 5: Stability scores under a fixed computational budget (number of histogram insertions). MABSplit allows more trees to be trained, which leads to greater feature stabilities across the forests.

adaptivity to the data distribution. Indeed, robustness to choice of hyperparameters is one of primary appeals of algorithms like RF [50].

MABSplit avoids the expensive $O(n\log n)$ sort used in many existing baselines and the $O(n)$ computational complexity of their corresponding histogrammed versions. MABSplit can be used in conjunction with existing software- and hardware-specific optimizations and with other methods such as Logarithmic Split-Point Sampling and boosting [61]. In boosting, MABSplit has the potential advantage of only needing to update data points' targets on-the-fly, as needed by its sampling, as opposed to current approaches that update targets for the entire dataset at each iteration. Additionally, MABSplit may permit easier parallelization due to lower memory requirements than existing algorithms, which may enable greater use in edge computing and may be adaptable to streaming settings.

We also note that $\log n^2$ term in the complexity of MABSplit (Equation 7) comes from a union bound over all possible $n^2$ confidence intervals computed for each feature-threshold pair across all stages of the algorithm (see the proof of Theorem 1 in Appendix 1). This union bound may be weak. Instead, one can more precisely compute the number of confidence intervals in terms of the arm gaps (the $\Delta_i$'s). In this way, the complexity result (Equation 7) can be phrased in terms of the $\Delta_i$'s instead of $n$. Intuitively, this would lead to a complexity bound in terms of the data-generating distribution that is independent of dataset size. Indeed, this is how complexity results are usually stated for multi-armed bandit algorithms [31]. We leave a more detailed investigation of this topic to future work.

**Acknowledgements:** M. T. was funded by a J.P. Morgan AI Fellowship, a Stanford Indisciplinary Graduate Fellowship, a Stanford Data Science Scholarship, and an Oak Ridge Institute for Science and Engineering Fellowship. M.J.Z. is supported by NIH grant R01 MH115676. I. S. was supported in part by the National Science Foundation (NSF) under grant CCF-2046991.

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
