# Appendix

## 1 Proofs

In this section, we present the proof of Theorem 1.

*Proof.* Following the multi-armed bandit literature, we refer to each feature-threshold pair $(f, t)$ as an arm and refer to its optimization objective $\mu_{ft}$ as the arm parameter. Pulling an arm corresponds to evaluating the change in impurity induced by one data point at one feature-threshold pair $(f, t)$ (i.e., arm) and incurs an $O(1)$ computation. This allows us to focus on the number of arm pulls, which translates directly to sample complexity.

First, we show that, with probability at least $1 - \frac{1}{n}$, all confidence intervals computed throughout the algorithm are valid, in that they contain the true parameter $\mu_{ft}$. For a fixed $(f, t)$ and a given iteration of the algorithm, the $(1 - \delta)$ confidence interval satisfies

$$\Pr\left(|\mu_{ft} - \hat{\mu}_{ft}| > C_{ft}\right) \le \delta.$$

Let $B$ denote the batch size chosen for MABSplit. Note that there are at most $\frac{n}{B}$ rounds in the main `while` loop (Line 6) of Algorithm 1 and hence at most $\frac{nmT}{B} \le nmT$ confidence intervals computed across all arms and all steps of the algorithm. With $\delta = \frac{1}{n^2 mT}$, we see that $\mu_{ft} \in [\hat{\mu}_{ft} - C_{ft}, \hat{\mu}_{ft} + C_{ft}]$ for every arm $(f, t)$ and for every step of the algorithm with probability at least $1 - \frac{1}{n}$, by a union bound over at most $nmT$ confidence intervals.

Next, we prove the correctness of Algorithm 1. Let $(f^*, t^*) = \arg\min_{f \in \mathcal{F}, t \in \mathcal{T}_f} \mu_{ft}$ be the desired output of the algorithm. Since the main `while` loop in the algorithm can only run $\frac{n}{B}$ times, the algorithm must terminate. Furthermore, if all confidence intervals throughout the algorithm are correct, it is impossible for $(f^*, t^*)$ to be removed from the set of candidate arms. Hence, $(f^*, t^*)$ (or some $(f, t)$ with $\mu_{ft} = \mu_{f^*t^*}$) must be returned upon termination with probability at least $1 - \frac{1}{n}$. This proves the correctness of Algorithm 1.

Finally, we consider the complexity of Algorithm 1. Let $n_{\text{used}}$ be the total number of arm pulls computed for each arm remaining in the set of candidate arms at a given point in the algorithm. Notice that, for any suboptimal arm $(f, t) \ne (f^*, t^*)$ that has not left the set of candidate arms, we must have $C_{ft} \le c_0 \sqrt{\frac{\log 1/\delta}{n_{\text{used}}}}$ by assumption. With $\delta = \frac{1}{n^2 mT}$ as above and $\Delta_{ft} = \mu_{ft} - \mu_{f^*t^*}$, if $n_{\text{used}} > \frac{4c_0^2}{\Delta_{ft}^2} \log(n^2 mT)$ then

$$2(C_{ft} + C_{f^*t^*}) \le 2c_0 \sqrt{\log(n^2 mT)/n_{\text{used}}} < \Delta_{ft} = \mu_{ft} - \mu_{f^*t^*},$$

and

$$\begin{aligned}
\hat{\mu}_{ft} - C_{ft} &> \mu_{ft} - 2C_{ft} \\
&= \mu_{f^*t^*} + \Delta_{ft} - 2C_{ft} \\
&\ge \mu_{f^*t^*} + 2C_{f^*t^*} \\
&> \hat{\mu}_{f^*t^*} + C_{f^*t^*}
\end{aligned}$$

which means that $(f, t)$ must be removed from the set of candidate arms at the end of that iteration. Hence, the number of data point computations $M_{ft}$ required for any arm $(f, t) \ne (f^*, t^*)$ is at most

$$M_{ft} \le \min\left[\frac{4c_0^2}{\Delta_{ft}^2} \log(n^2 mT) + B, 2n\right].$$

Notice that this holds simultaneously for all arms $(f, t)$ with probability at least $1 - \frac{1}{n}$. We conclude that the total number of arm pulls $M$ satisfies

$$\mathbb{E}[M] \le \mathbb{E}[M \,|\, \text{all confidence intervals are correct}] + \frac{1}{n}(2nMT)$$

$$\le \sum_{f \in \mathcal{F}, t \in \mathcal{T}_f} \min\left[\frac{4c_0^2}{\Delta_{ft}^2} \log(n^2 mT) + B, 2n\right] + 2mT,$$

where we used the fact that the maximum number of computations for any arm is $2n$. As argued before, since each arm pull involves an $O(1)$ computation, $M$ also corresponds the total number of computations. $\square$

## 2 $O(\log n)$ **scaling of MABSplit**

In Theorem 1, we demonstrated that MABSplit scales logarithmically in dataset size. In this section, we empirically validate this claim.

Appendix Figure 1 (a) demonstrates the number of data points queried by MABSplit for a single node split, i.e., a single call to MABSplit, as the dataset size increases, for various subset sizes of MNIST. For each sample size, a sample is drawn with replacement from the original MNIST dataset. The model is trained using Gini impurity in for the usual digitr classification task.

Appendix Figure 1 (b) demonstrates the same plot for various subset sizes of the Random Linear Model dataset. The Random Linear Model dataset consists of 200,000 datapoints with 50 features, 6 of which are correlated with the targets and 44 of which are pure noise, using `scikit-learn`'s `make_regression` function [47]; 160,000 datapoints are used for training and the remaining 40,000 for test.

Appendix Figure 1 also shows the best linear and logarithmic fits to each dataset. The relatively high $R^2$ values of the logarithmic fits ($R^2 = 0.97$ and $R^2 = 0.82$) compared to that of the linear fits ($R^2 = 0.66$ and $R^2 = 0.43$) suggests that the scaling of MABSplit is logarithmic (and therefore sublinear) with dataset size.

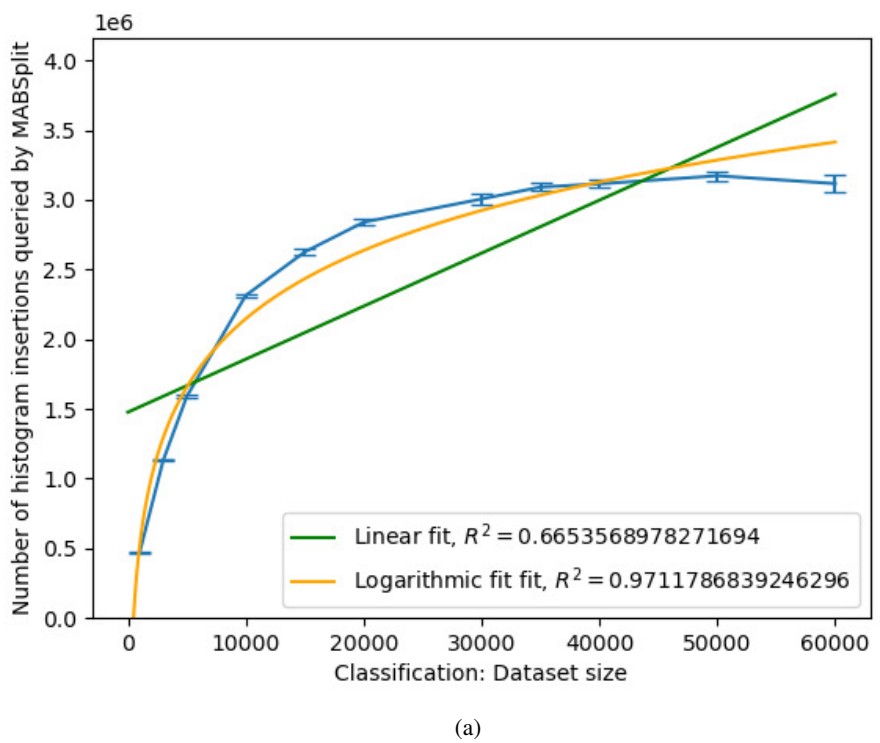

(a)

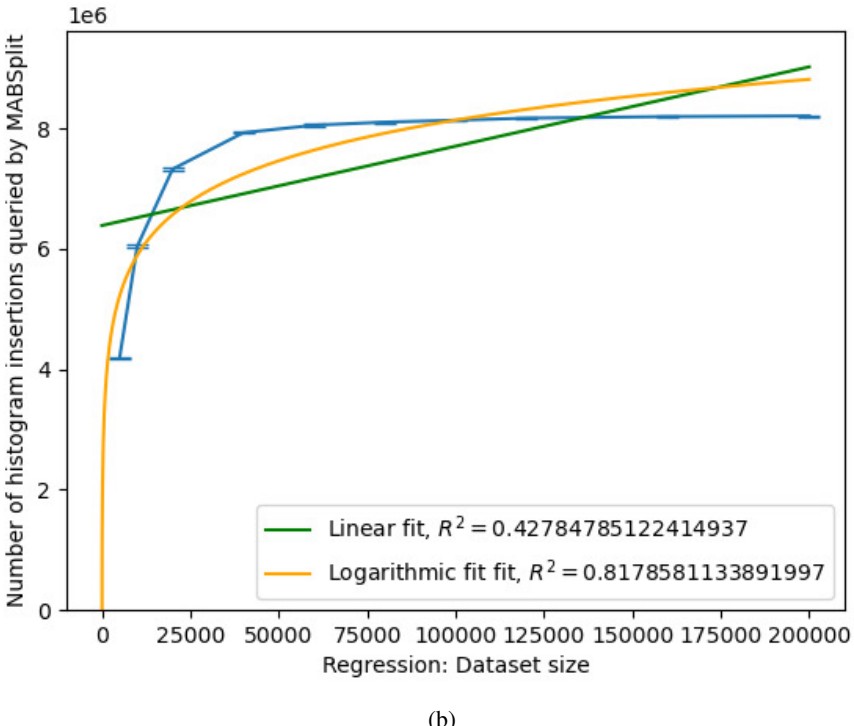

(b)

Appendix Figure 1: Scaling of MABSplit with different data subset sizes for (a) the MNIST digit classification task and (b) the Random Linear Model regession task. In both tasks, MABSplit appears to scale logarithmically, not linearly, with dataset size.

# 3 Mean Estimation and Confidence Interval Constructions

In this section, we discuss the estimation of the means $\mu_{ft}$ and construction of their confidence intervals via plug-in estimators and the delta method.

Let $p_{\text{L},k}$, $p_{\text{R},k}$, $\hat{p}_{\text{L},k}$, and $\hat{p}_{\text{R},k}$ be the same as defined in Subsection 3.1. Furthermore, let $\mathbf{p} = [p_{\text{L},1}, \cdots, p_{\text{L},K}, p_{\text{R},1}, \cdots, p_{\text{R},K}]^T$ and $\hat{\mathbf{p}} = [\hat{p}_{\text{L},1}, \cdots, \hat{p}_{\text{L},K}, \hat{p}_{\text{R},1}, \cdots, \hat{p}_{\text{R},K}]^T$. Then, $n'\hat{\mathbf{p}}$ follows a multinomial distribution with parameters $(n', 2K, \mathbf{p})$.

Let $\boldsymbol{\theta} = [p_{\text{L},1}, \cdots, p_{\text{L},K}, p_{\text{R},1}, \cdots, p_{\text{R},K-1}]^T$ and $\hat{\boldsymbol{\theta}} = [\hat{p}_{\text{L},1}, \cdots, \hat{p}_{\text{L},K-1}, \hat{p}_{\text{R},1}, \cdots, \hat{p}_{\text{R},K-1}]^T$. Then, by the Central Limit Theorem,

$$\sqrt{n'}(\hat{\boldsymbol{\theta}} - \boldsymbol{\theta}) \overset{D}{\sim} \mathcal{N}(0, \Sigma), \tag{8}$$

where $\Sigma_{ii} = \theta_i(1 - \theta_i)$ and $\Sigma_{ij} = -\theta_i\theta_j$.

Next, we write $\mu_{ft}$ in terms of $\boldsymbol{\theta}$ for the impurity metrics as

$$\text{Gini impurity}: \ \mu_{ft}(\boldsymbol{\theta}) = 1 - \frac{\sum_{k=1}^{K} \theta_k^2}{\sum_{k=1}^{K} \theta_k} - \frac{\sum_{k=K+1}^{2K-1} \theta_k^2 + (1 - \sum_{k=1}^{2K-1} \theta_k)^2}{1 - \sum_{k=1}^{K} \theta_k}, \tag{9}$$

$$\text{Entropy}: \ \mu_{ft}(\boldsymbol{\theta}) = -\sum_{k=1}^{K} \theta_k \log_2 \frac{\theta_k}{\sum_{k'=1}^{K} \theta'_k} - \sum_{k=K+1}^{2K-1} \theta_k \log_2 \frac{\theta_k}{1 - \sum_{k'=1}^{K} \theta'_k} -$$

$$(1 - \sum_{k=1}^{2K-1} \theta_k) \log_2 \frac{(1 - \sum_{k=1}^{2K-1} \theta_k)}{1 - \sum_{k=1}^{K} \theta_k}. \tag{10}$$

For a given impurity metric, let $\nabla\mu_{ft}(\boldsymbol{\theta})$ be the derivative of $\mu_{ft}$ with respect to $\boldsymbol{\theta}$. From the delta method,

$$\sqrt{n'}(\hat{\mu}_{ft}(\boldsymbol{\theta}) - \mu_{ft}(\boldsymbol{\theta})) \overset{D}{\sim} \mathcal{N}(0, \nabla\mu_{ft}(\boldsymbol{\theta})^T \Sigma \nabla\mu_{ft}(\boldsymbol{\theta})), \tag{11}$$

where the CIs can be constructed accordingly. These CIs are asymptotically valid as $n', n \to \infty$. For other impurity metrics such as MSE, the CIs can be similarly derived by writing the corresponding $\mu_{ft}$ in terms of $\boldsymbol{\theta}$ and computing $\nabla\mu_{ft}(\boldsymbol{\theta})$.

| Model | Task and Dataset | Performance Metric | Test Performance |
|---|---|---|---|
| RF (ours) | Classification: 20 Newsgroups | Accuracy | $74.1 \pm 2.8\%$ |
| RF (scikit-learn) | Classification: 20 Newsgroups | Accuracy | $76.2 \pm 1.7\%$ |
| ExtraTrees (ours) | Classification: 20 Newsgroups | Accuracy | $66.5 \pm 5.1\%$ |
| ExtraTrees (scikit-learn) | Classification: 20 Newsgroups | Accuracy | $62.6 \pm 2.8\%$ |
| RF (ours) | Regression: California Housing | MSE | $0.679 \pm 0.022$ |
| RF (scikit-learn) | Regression: California Housing | MSE | $0.672 \pm 0.028$ |
| ExtraTrees (ours) | Regression: California Housing | MSE | $0.696 \pm 0.055$ |
| ExtraTrees (scikit-learn) | Regression: California Housing | MSE | $0.695 \pm 0.082$ |

Appendix Table 1: Comparison of our re-implementation of baselines with the the implementations available in scikit-learn. No statistically significant differences are apparent, which suggests that our re-implementations are accurate.

## 4    Comparison of baseline implementations and scikit-learn

In this section, we compare our re-implementation of common baselines to those in popular packages to verify the accuracy of our re-implementation. Specifically, we compare our implementations of Random Forest Classifiers, Random Forest Regressors, Extremely Random Forest Classifiers, and Extremely Random Forest Regressors to those of scikit-learn. We omit comparisons of the Random Patches models because their correctness is implied by that of the Random Forest model, as the Random Patches model consists of applying the Random Forest model to subsampled data and features.

For classification, we compare our implementations on the 20 newsgroups dataset filtered to two newsgroups, alt.atheism and sci.space. The dataset is embedded via TF-IDF and projected onto their top 100 principal components, following standard practice [47]. The train-test split is the standard one provided by scikit-learn.

For all classification problems, we average the predicted probabilities of each tree in the forest ("soft voting") as opposed to only allowing each tree to vote for a single class ("hard voting"), following the implementation in scikit-learn [47].

For regression, we compare our implementations on the California Housing dataset, subsampled to 1,000 points as performing the regression on the full dataset of approximately 20,000 points is computationally prohibitive. The train-test split is the standard one provided by scikit-learn [47].

Table 1 presents our results. In all cases, our re-implemented baselines do not present a statistically significant difference in performance from the models present in scikit-learn, which suggests that our re-implementations are correct. Performance is measured over 20 random seeds to compute averages and standard deviations.

# 5 Profiles

In this work, we focused on the reducing the runtime at the *algorithmic* level, i.e., reducing the complexity of computing the best feature-threshold split. In this section, we justify this choice by demonstrating that most of the time spent in our re-implementation of the baseline algorithms is spent in computing the best feature-threshold split.

Appendix Figure 2 demonstrates the wall-clock time spent inside various functions when fitting a Random Forest classifier without MABSplit on two subsets of the MNIST dataset of sizes 5,000 and 10,000. Most of the time is spent inside the computation of the best feature-threshold split, which scales approximately as dataset size and motivates our focus on improving the performance of the split-identification subroutine. When using MABSplit, the time spent to identify the best feature-threshold split is reduced significantly (Appendix Figure 3).

Appendix Figure 4 also contains an example callgraph demonstrating callers and callees for the fitting procedure of a Random Forest, for easier interpretation of Appendix Figures 2 and 3.

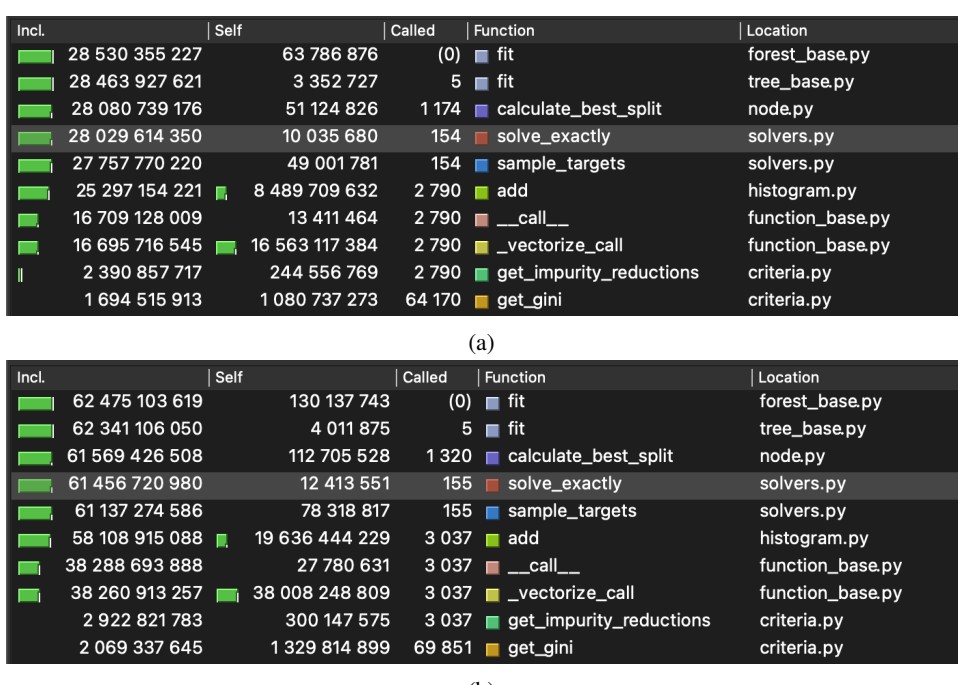

(a)

(b)

Appendix Figure 2: Profiles for the node-splitting algorithm using the exact solver/naïve computation, the canonical algorithm for computing the best feature-threshold split, for 5,000 (top) and 10,000 (bottom) data point subsets of MNIST. The "Function" column is the name of the called function, the "Incl." column is the time spent in the function and any called subroutines, and the "Self" column is the time (in nanoseconds) spent in only the function and *not* in any callees. All times are in nanoseconds. When increasing the dataset size, the overhead spent outside of the `solve_exactly` function grows negligibly from about 0.5 seconds to about 1 second. However, the time spent in the `solve_exactly` function and any called subroutines grows from about 28 seconds to about 61 seconds and constitutes approximately 98% of the increase in wall-clock time. This observation motivates our focus on improving the subroutine used to identify the best feature-threshold split. This profile was generated with `cProfile` and visualized with `pyprof2calltree` [37].

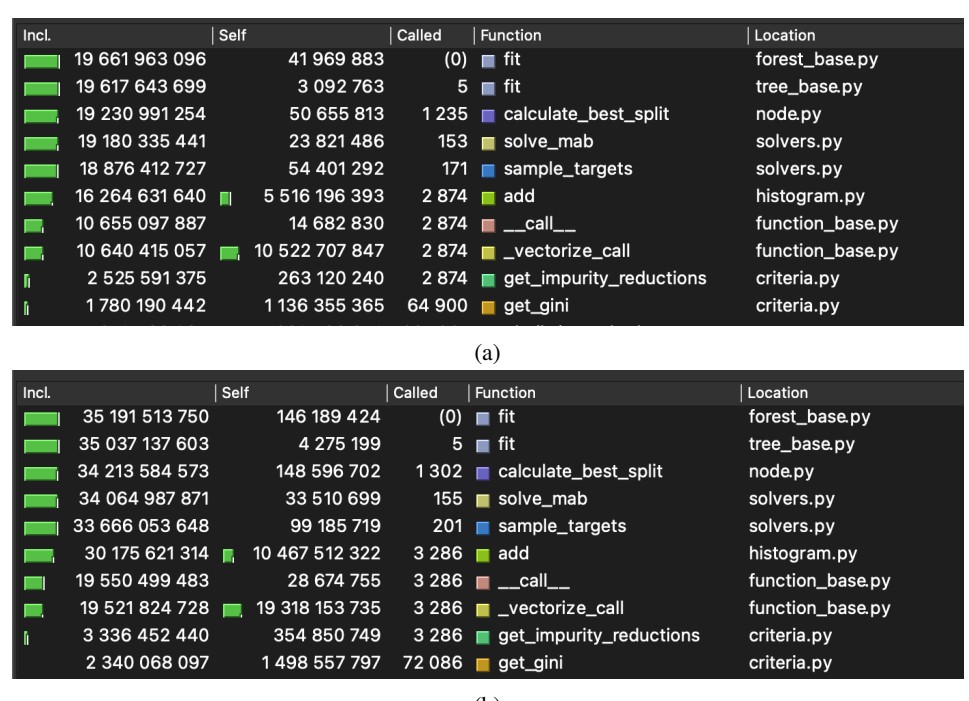

| Incl. | | Self | Called | Function | Location |
|---|---|---|---|---|---|
| | 19 661 963 096 | 41 969 883 | (0) | fit | forest_base.py |
| | 19 617 643 699 | 3 092 763 | 5 | fit | tree_base.py |
| | 19 230 991 254 | 50 655 813 | 1 235 | calculate_best_split | node.py |
| | 19 180 335 441 | 23 821 486 | 153 | solve_mab | solvers.py |
| | 18 876 412 727 | 54 401 292 | 171 | sample_targets | solvers.py |
| | 16 264 631 640 | 5 516 196 393 | 2 874 | add | histogram.py |
| | 10 655 097 887 | 14 682 830 | 2 874 | __call__ | function_base.py |
| | 10 640 415 057 | 10 522 707 847 | 2 874 | _vectorize_call | function_base.py |
| | 2 525 591 375 | 263 120 240 | 2 874 | get_impurity_reductions | criteria.py |
| | 1 780 190 442 | 1 136 355 365 | 64 900 | get_gini | criteria.py |

(a)

| Incl. | | Self | Called | Function | Location |
|---|---|---|---|---|---|
| | 35 191 513 750 | 146 189 424 | (0) | fit | forest_base.py |
| | 35 037 137 603 | 4 275 199 | 5 | fit | tree_base.py |
| | 34 213 584 573 | 148 596 702 | 1 302 | calculate_best_split | node.py |
| | 34 064 987 871 | 33 510 699 | 155 | solve_mab | solvers.py |
| | 33 666 053 648 | 99 185 719 | 201 | sample_targets | solvers.py |
| | 30 175 621 314 | 10 467 512 322 | 3 286 | add | histogram.py |
| | 19 550 499 483 | 28 674 755 | 3 286 | __call__ | function_base.py |
| | 19 521 824 728 | 19 318 153 735 | 3 286 | _vectorize_call | function_base.py |
| | 3 336 452 440 | 354 850 749 | 3 286 | get_impurity_reductions | criteria.py |
| | 2 340 068 097 | 1 498 557 797 | 72 086 | get_gini | criteria.py |

(b)

Appendix Figure 3: Profiles for the node-splitting algorithm using MABSplit, for 5,000 (top) and 10,000 (bottom) datapoint subsets of MNIST. The "Function" column is the name of the called function, the "Incl." column is the time spent in the function and any called subroutines, and the "Self" column is the time (in nanoseconds) spent in only the function and *not* in any called subroutines. All times are in nanoseconds. When increasing the dataset size, the time spent in the `solve_mab` function and any called subroutines only grows from approximately 20 seconds to approximately 35 seconds to identify the best feature-threshold split. This profile was generated with `cProfile` and visualized with `pyprof2calltree` [37].

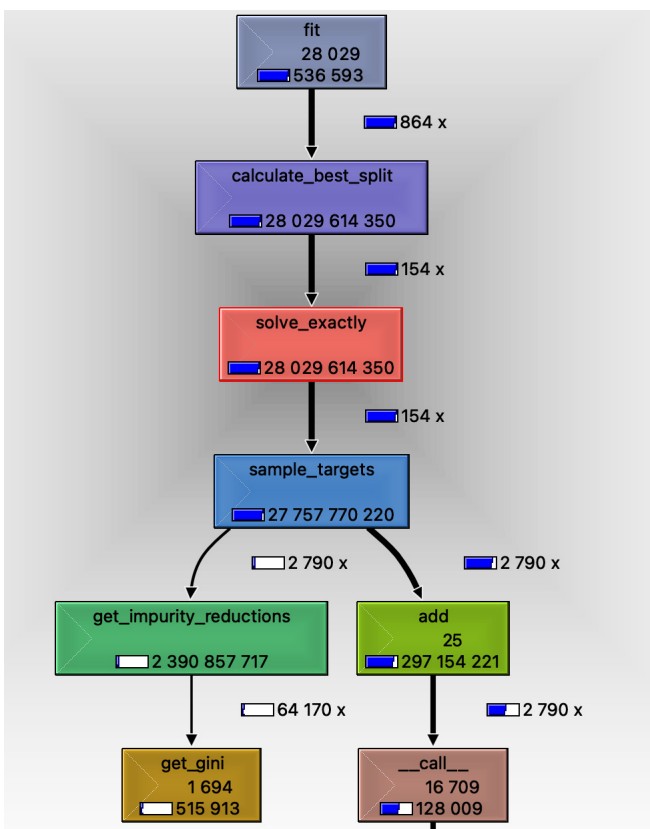

Appendix Figure 4: Example call graph of the `fit` subroutine for the forest-based models in our re-implementation when the forest includes a single tree to be split only once. The `fit` method of the forest calls the `fit` method of its only tree, which calls `calculate_best_split` method of the root node, which calls the respective solver (`solve_exactly` for the brute-force algorithm or `solve_exactly` for MABSplit), where the majority of wall-clock time is spent.

# 6 Experiment Details

Here we provide full details for the experiments in Section 5. All experiments were run on 2021 MacBook Pro running MacOS 12.5.1 (Monterey) with an Apple M1 Max processor, and 64 GB RAM.

## 6.1 Datasets

**Classification Datasets:** We use the MNIST [38], APS Failure at Scania Trucks [25, 20], and Forest Cover Type [13, 20] datasets. The MNIST dataset consists of 60,000 training and 10,000 test images of handwritten digits, where each black-and-white image is represented as a 784-dimensional vector and the task is to predict the digit represented by the image. The APS Failure at Scania Trucks dataset consists of 60,000 datapoints with 171 features and the task is to predict component failure. The Forest Covertype dataset consists of 581,012 datapoints with 54 feature and the task is to predict the type the forest cover type from cartographic variables.

**Regression Datasets:** We use the Beijing Multi-Site Air-Quality [64, 20] and the SGEMM GPU Kernel Performance [8, 44, 20] datasets. The Beijing Multi-Site Air-Quality dataset consists of 420,768 datapoints with 18 features and the task is to predict the level of air pollution. The SGEMM GPU Kernel Performance dataset consists of 241,600 datapoints and the task is to predict the running time of a matrix multiplication.

For all datasets except MNIST (which has predefined training and test datasets), all datasets were randomized into 9:1 train-test splits. All datasets are publicly available.

## 6.2 Runtime Experiments

For the runtime experiments presented in Tables 1, all performances were measured from 5 random seeds. For all datasets, the maximum depth was set to 1 except for the MNIST dataset, in which the maximum depth was set to 5. The number of trees in each model was set to 5. All experiments used the Gini impurity criterion and the minimum impurity decrease required from performing a split was set to 0.005. For the Random Patches (RP) model, $\alpha_n$ was set to 0.7 and $\alpha_f$ was set to 0.85.

For the regression runtime experiments presented in Table 2, all performances were measured from 5 random seeds. For the Beijing Multi-Site Air-Quality Dataset, the maximum depth was set to 1 and for the SGEMM GPU Kernel Performance Dataset, the maximum number of leaf nodes was set to 5. The number of trees in each model was set to 5. All experiments used the MSE impurity criterion and the minimum impurity decrease required from performing a split was set to 0.005. For the Random Patches (RP) model, $\alpha_n$ was set to 0.7 and $\alpha_f$ was set to 0.85.

## 6.3 Budget Experiments

For the classification budget experiments presented in Table 3, all performances were measured from 5 random seeds. The budget for each model on the MNIST, APS Failure at Scania Trucks, and Forest Covertype datasets were set to 10,192,000, 784,000, and 9,408,000, respectively. For the Random Patches (RP) model, $\alpha_n$ was set to 0.6 and $\alpha_f$ was set to 0.8. The maximum number of trees in any model was set to 100 and the maximum depth of each tree was set to 5.

For the regression budget experiments presented in Table 4, all performances were measured from 5 random seeds. The budget for each model on the Beijing Multi-Site Air-Quality Dataset was set to 76,800,000 and the budget for each model on the SGEMM GPU Kernel Performance Dataset was set to 24,000,000. For the Random Patches (RP) model, $\alpha_n$ was set to 0.8 and $\alpha_f$ was set to 0.5. The maximum number of trees in any model was set to 100 and the maximum depth of each tree was set to 5.

## 6.4 Stability Experiments

Two metrics for calculating feature importance are used in Table 5: out-of-bag Permutation Importance (OOB PI) and Mean Decrease in Impurity (MDI) [42, 49]. For a feature $f$, the OOB PI is calculated by measuring the difference between the trained model's out-of-bag error on the original data with its out-of-bag error on all the data with all out-of-bag datapoints' $f$ values shuffled. The

MDI for a feature $f$ is the average decrease in impurity of all nodes where $f$ is selected as the splitting criterion.

Once feature importances have been calculated, the top $k$ most important features for the model are selected and the stability of these $k$ features is measured via standard stability formulas [43].

The results of the stability experiments are shown in Table 5. The Random Classification dataset is generated via scikit-learn's `datasets.make_classification` function with `n_samples=10000`, `n_features=60`, and `n_informative=5`. The Random Regression dataset is generated by scikit-learn's `datasets.make_regression` with `n_samples=10000`, `n_features=100`, and `n_informative=5`.

| MNIST Dataset (Classification, $N = 60,000$, maximum depth $= 8$) | | |
|---|---|---|
| Model | Wall-clock Training Time (s) | Accuracy (%) |
| `scikit-learn` Decision Tree Classifier | 34.665 ± 1.266 | 91.061 ± 0.0 |
| Histogrammed decision tree (Exact solver, ours) | 86.514 ± 2.839 | 90.923 ± 0.0 |
| **Histogrammed decision tree (MABSplit solver, ours)** | **8.538 ± 0.079** | **90.629 ± 0.234** |

Appendix Table 2: Comparison of accuracy and wall-clock training time of `scikit-learn`'s Decision Tree Classifier with our implementation on the MNIST digit classification task. Our implementation of the histogrammed decision tree is slower than `scikit-learn`'s, but our optimized implementation is about 4x faster than `scikit-learn`'s. The slight performance degradation is likely due to discretization of the data during histogramming; this effect is also seen when histogramming the data and using the exact solver (i.e., when not using MABSplit). A more heavily optimized version of our histogrammed decision tree when using MABSplit would likely result in even lower training times. Performance was measured over 5 random seeds.

# 7 Limitations

## 7.1 Theoretical Limitations

Crucial to the success of MABSplit are the assumptions described before and after Theorem 1. In particular, we assume that their is reasonable heterogeneity amongst the true impurity reductions of different feature-value splits. Such assumptions are common in the literature and have been validated on many real-world datasets [5, 63, 7, 58, 4, 6].

We also note that the assumptions that each CI scales as $\sqrt{\frac{\log 1/\delta}{n'}}$ may be violated when using certain impurity metrics. For example, the derivative of the entropy impurity criterion with respect to some $p_k$ approaches $\infty$ when $p_k \rightarrow 0$. In this case, we cannot apply the delta method from Appendix 3 to compute finite CIs that scale in the way we require. In such settings, it may be necessary to compute the CIs in other ways, e.g., following [46] or [9].

We note that in the worst case, even when all assumptions are violated, MABSplit is never worse than the naïve algorithm in terms of sample complexity. In the worst case, it is a batched version of the naïve algorithm.

## 7.2 Practical Limitations

We note that MABSplit may perform worse than naïve node-splitting on very small datasets, where the overhead of sampling the data in batches outweighs any potential benefits in sample complexity (see Appendix 8 for further discussion).

In this work, we avoided a direct runtime comparison with `scikit-learn` because `scikit-learn` utilizes a number of low-level implementation optimizations that would make the comparison unfair. To provide a brief comparison to the popular `scikit-learn` implementation, however, we attempted to optimize our implementation using `numba` [36], a package that translates Python code to optimized machine code. Our `numba`-optimized implementation is 4x faster than `scikit-learn`'s `DecisionTreeClassifier` and achieves comparable performance on the MNIST dataset; see Appendix Table 2.

In order for practitioners to take full advantage of MABSplit, however, it may be necessary to implement MABSplit within the `scikit-learn` library. In doing so, it may be possible that MABSplit makes it difficult or impossible to use existing optimizations in the `scikit-learn` library. An example of this is vectorization: because the naïve node-splitting algorithm queries the data in a predictable way, each datapoint can be queried more quickly than in MABSplit. Despite MABSplit's advantages in sample complexity, the disadvantages of being unable to use implementation optimizations like vectorization may outweigh MABSplit's benefits. Many of these risks may be ameliorated by addressed MABSplit into existing RF implementations such as the one in `scikit-learn`. We anticipate that many optimizations will still apply: for example pre-fetching data to have it in caches close to the CPU, manual loop unrolling, etc. We leave an optimization implementation of MABSplit inside the `scikit-learn` library to future work.

# 8   Comparison on Small Datasets

In this section, we investigate the performance of MABSplit on small datasets. Appendix Figure 5 demonstrates the performance of MABSplit, both in wall-clock training time and sample complexity, for various subset sizes of MNIST. Our results that RF+MABSplit outperforms the standard RF algorithm, in both sample complexity and wall-clock time, when the dataset size exceeds approximately 1100 datapoints.

However, we also note that the main use case for MABSplit is when the data size is large and it is computationally challenging to run standard forest-based algorithms. Indeed, the use of big data in many applications that necessitate sampling was the primary motivation for our work [11, 54, 18, 57, 41, 1].

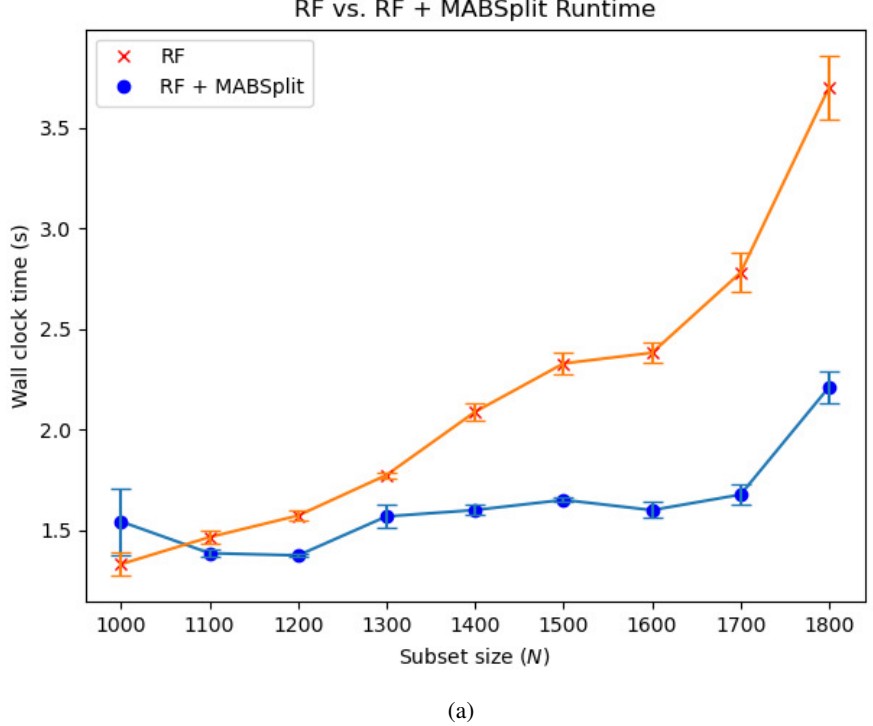

(a)

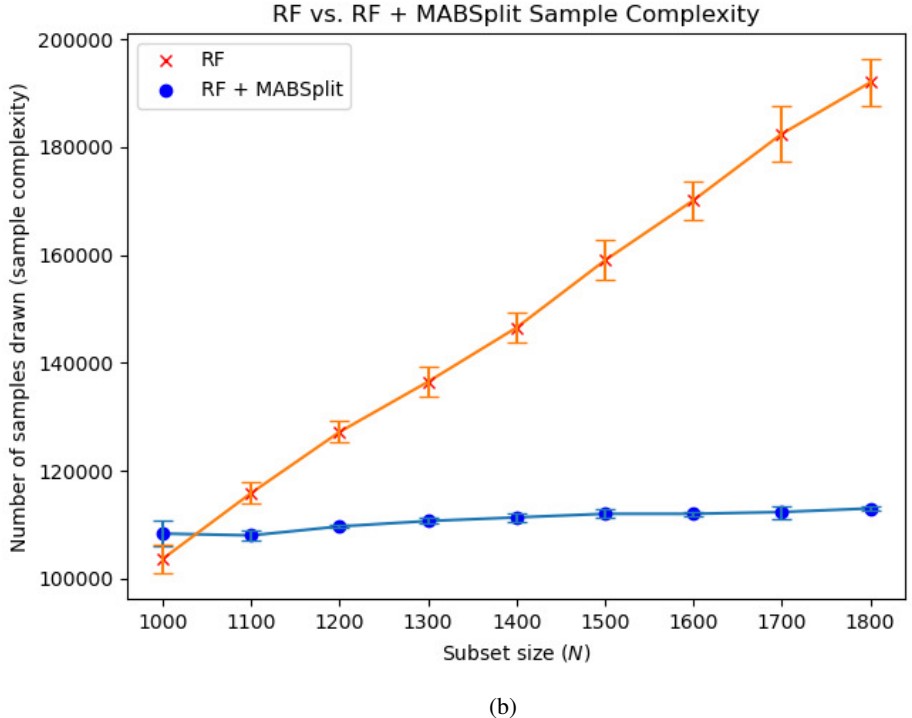

(b)

Appendix Figure 5: (a) Wall-clock training times and (b) sample complexities of a random forest model with and without MABSPlit, for various subset sizes of MNIST. For dataset sizes below approximately 1000, the exact random forest model performs better in terms of sample complexity and wall-clock time. Above 1100 datapoints, the MABSplit version demonstrates better sample complexity and wall-clock time. Error bars were computed over 3 random seeds. Test performances were not different at a statistically significant level.

# 9 Description of Other Node-Splitting Algorithms

For completeness, we provide a brief description of various baseline models' node-splitting algorithms here to enable easier comparison with MABSplit.

Consider a node with $n$ datapoints each with $m$ features, and $T$ possible thresholds at which to split each feature. We discuss the classification setting for simplicity, though the same arguments apply to regression.

A very naïve approach would be to iterate over all $mT$ feature-value splits, and compute the probabilities $p_{L,k}$ and $p_{R,k}$ from all $n$ datapoints. This results in complexity $O(mTn)$, which is $O(mn^2)$ when $T = n$ (for example, $T = n$ in the un-histogrammed setting).

Instead, the usual RF algorithm sorts all $n$ datapoints in $O(n\log n)$ time for each of the $m$ features, resulting in total computational cost $O(mn\log n)$. Then the algorithm scans linearly from lowest value to highest value for each feature and update the parameters $p_{L,k}$ and $p_{R,k}$ via simple counting to find the best impurity reduction for each of the $T$ potential splits. The complexity of this step is $O(mT + mn)$, where the "$+mn$" comes from the allocations of each data point to the left or right node during the scan (each data point is re-allocated only once per feature). Thus the total complexity of this approach is $O(mn\log n + mT + mn) = O(mn\log n + mT)$. This is $O(mn\log n)$ when $T = n$.

The binned (a.k.a. histogrammed) method does not require the per-feature sort and avoids the $O(mn\log n)$ computation when $T < n$,. Instead, each of the $n$ points must be inserted into the correct bin (which can be done in $O(1)$ time for each datapoint if the bins are equally spaced) for each of the $m$ features, incurring total computational cost $O(mn)$. Then, the same linear scanning approach as in the "standard" algorithm is performed with complexity $O(mT + mn)$. The total complexity of this approach is $O(mn + mT + mn) = O(m(n + T))$. This is $O(mn)$ when $T = n$.

In general, we do not assume $T = n$, i.e., that every feature value is a potential split point, unless otherwise specified. In our paper, the "standard" approach refers to the **un**binned approach which requires an $O(mn\log n)$ sort and "linear" refers to the binned approach that is $O(m(n + T))$, which is $O(mn)$ when $T = O(n)$.

Crucially, when $T = o(N)$ (as is often the case in practice, e.g., for a constant number of bins) and the necessary gap assumptions are satisfied, MABSplit scales as $O(mT\log n)$. In many cases, this is much better than $O(m(n + T))$, e.g., for large datasets, because the dependence on $n$ is reduced from linear to logarithmic. More concretely, treating $T$ as a constant and ignoring the dependence on $m$, we reduce the complexity of the binned algorithm from $O(n)$, what we refer to as "linear," to $O(\log n)$.