# OpenReview forum: "MABSplit: Faster Forest Training Using Multi-Armed Bandits"
_NeurIPS.cc/2022/Conference — NeurIPS 2022 Accept_

### Official Review · Reviewer_AzdZ · 2022-07-11

**Rating:** 6
**Confidence:** 4
**Soundness:** 4 excellent
**Presentation:** 3 good
**Contribution:** 3 good

**Summary:**

The paper "MABSplit: Faster Forest Training Using Multi-Armed Bandits" proposes to estimate the performance of the splits during DT induction instead of evaluating them exactly. To do so, it views the splitting procedure as a multi-armed bandit problem which enables confidence bounds for each split that in-turn can be used to eliminate unpromising features "early on" by evaluating them only on a small subset of data points.

**Questions:**

1) You keep referring to a linear runtime for the exact split computation and I think I understand how you mean this, but please clarify the following for me: A very naive solution of finding the exact split is to iterate over all feature / split pairs (of which there are at most $Nm$ where $m$ is the number of features) and then to iterate over all data points to compute the probabilities $p_{L,k}$ and $p_{R,k}$ . A more evolved approach (which is implemented in most frameworks) would first sort the data points in $\mathcal O(dN \log N)$ and then perform two linear scans in which the statistics for the left and the right node are updated. Last, we can perform binning of the feature values into histograms which then requires a linear scan through the data. The exact procedure and runtime depends on the binning method. Assuming one feature ($m=1$) we have:
	1) The very naive approach has $\mathcal O(N^2)$ runtime.
	2) The standard approach has $\mathcal O(N \log N + 2N) = \mathcal O(N \log N)$ runtime
	3) The binning method has $\mathcal O(N)$ plus some additional overhead for the exact binning method.
	4) You refer to "linear runtime" to the linear scan in binning and not the linear scan in the "standard approach". This is confusing because you basically always talk about binning, but then mention sorting in line 306 (in the conclusion).
2) You claim that MABSplit has a (sub-)linear runtime and I understand your remark under Theorem 1. However, just to clarify: The worst case runtime can still be $\mathcal O(N^2)$ when we have $|\mathcal F| \cdot |\mathcal T_f|  \approx N$, correct?

**Limitations:**

The expected runtime is analysed, but there is no worst-case runtime presented (see my question). Personally, this would improve the paper, but I don't think it absolutely necessary for acceptance.
The experimental evaluation is limited in a sense that they use a simple, non-optimized baseline. However, this is explicitly stated and makes sense in the context of the paper.


**Strengths And Weaknesses:**

The paper has a solid theoretical background and a novel algorithmic contribution, but lacks a thorough experimental evaluation.
- [+] Novel and non-trivial extension of the node-splitting in DTs
- [+] Solid theoretical work
- [+] Solid literature review
- [-] Experimental evaluation can be extended
- [-] Wording of the paper is sometimes unclear.

# Detailed Review
I  like the idea and find it really interesting. I do not think I came across such a combination of bandit learning and DT induction yet and as such I find it a valuable contribution. While the authors do not explicitly motivate their work, the benefit should be clear to the reader. The related work section is solid, although I was a bit confused that XGBoost/LightGBM/CatBoost are not mentioned explicitly, although they are cited multiple times in the paper [12,24] . Section 2 and the first part of section 3 were also easy to follow in my opinion. Starting with Algorithm 1 it gets a bit troublesome. To me, the runtime of the proposed method is somewhat unclear. The authors give a theorem with the expected runtime, however, there is no worst-case runtime? I think the worst-case runtime is at-least also quadratic which means that for certain use-cases the proposed method is worse than the standard approach implemented in many frameworks (see my question below). Additionally, the  authors refer to a "linear" runtime multiple times in the paper, but I don't know exactly what they mean by this. Last, I find the experimental evaluation weak. It makes perfect sense to compare the algorithmic contribution to a base-line implementation, although implementing MABSplit into Weka or Scikit-learn would certainly improve the paper. However, the method is practically only evaluated on synthetic data and MNSIT. While using synthetic data is very important to isolate certain effects (e.g. the effect of the number and type of features is not discussed at all in the paper), some real-world experiments are also important. Last, and more crucial to me, the authors do not compare their approach against existing methods. More specifically, they do not compare it against the estimations used by Hoeffding's Trees [13,30] or the F-Forest [16] although these seem to be the only to similar methods available. I think a comparison here is urgently required.

## Things to improve if the paper is rejected
- Please refine what you mean by "linear runtime" throughout the paper. It would probably best to just give some pseudo-code of the exact split induction (with or without binning) to which you compare against. If space allows put that directly in the paper, otherwise use the appendix.
- Point out the worst-case runtimes and the expected runtime of your method and related work. It should be clear that you win in terms of expected runtime and have better empirical results than the standard that is implemented in Scikit-learn/Weka etc.
- Compare against Hoeffding Trees and/or F-Forest. As you write, these methods are very related so that you cannot ignore them in your evaluation
- Add more (real-world) datasets. Personally, I think its fine to either go for classification or regression to simplify the analysis, but then use more datasets with varying amount of features and feature types.

---

> ### Author Response · Authors · 2022-08-02
> **Response to Reviewer AzdZ (3 of 3)**
>
> (Continued from above)
>
> $\textbf{Reviewer AzdZ Comment 3:}$ Last, I find the experimental evaluation weak... the method is practically only evaluated on synthetic data and [MNIST]... the effect of the number and type of features is not discussed at all in the paper... some real-world experiments are also important...Add more (real-world) datasets. Personally, I think its fine to either go for classification or regression to simplify the analysis, but then use more datasets with varying amount of features and feature types.
>
> $\textbf{Response to Reviewer AzdZ Comment 3:}$
>
> Thank you for your comments. Following the revierwer's suggestion, we have tested MABSplit's performance across three additional classification datasets and two additional regression datasets. The addenda to each of Tables 1, 2, 3, and 4 is presented in a comment to all reviewers and we reference them in the remainder of our response. We will add them to the final paper (though we have omitted it from the revision due to the original space limit of 8 pages).
>
> In the additional classification experiments, MABSplit demonstrates increasing benefit across the classification tasks as the dataset size increases, from approximately 10x speedups on the APS Failure Dataset ($N = 60,000$) to approximately 30-50x speedups on the Forest Cover Type Dataset ($N = 581,012$). MABSplit also demonstrates superiority over baselines in the additional regression experiments on the Beijing Air Quality Dataset and SGEMM GPU Kernel Performance Dataset.
>
> $\textbf{Reviewer AzdZ Comment 4:}$ I was a bit confused that XGBoost/LightGBM/CatBoost are not mentioned explicitly, although they are cited multiple times in the paper [12,24].
>
> $\textbf{Response to Reviewer AzdZ Comment 4:}$ We will mention these models by name in the final paper.
>
> $\textbf{Reviewer AzdZ Comment 5:}$ Implementing MABSplit into Weka or Scikit-learn would certainly improve the paper...The experimental evaluation is limited in a sense that they use a simple, non-optimized baseline. However, this is explicitly stated and makes sense in the context of the paper.
>
> $\textbf{Response to Reviewer AzdZ Comment 5:}$
>
> We agree it is important to integrate MABSplit into widely-used machine learning packages such as $\texttt{scikit-learn}$. To do this, we need to integrate my algorithm into the complex library of $\texttt{scikit-learn}$ and to perform a series of low-level optimizations such as cache optimizations and manual loop unrolling. This may be out of the scope of the current work and we leave it as an interesting direction for future work. However, we have added a discussion of these potential limitations to Appendix 7 of the revision.
>
> We also note that our strategy of reimplementing the baseline algorithms may provide a more fair evaluation of the algorithmic improvement of our method, instead of focusing on improvements due to careful low-level optimizations. We note that a similar approach focusing on sample complexity instead of wall-clock time has also been used in related works, such as NeurIPS '20 [40], AAAI '19 [Liu et al., "A Bandit Approach to Maximum Inner Product Search"], and AISTATS '18 [4].
>
> Please let us know if you have any further questions and/or comments.

---

> > ### Comment · Reviewer_AzdZ · 2022-08-08
> > **Response to Response to Reviewer AzdZ**
> >
> > I thank the authors for their time and effort. I think the additional experiments improve the paper significantly. I would love to see some of the explanations / comments on the linear runtime in the final paper, but I think all of my concerns have been addressed adequately.

---

> ### Author Response · Authors · 2022-08-02
> **Response to Reviewer AzdZ (2 of 3)**
>
> (Continued from above)
>
> Crucially, when $T = o(N)$ (as is often the case in practice, e.g., for a constant number of bins) and the necessary assumptions are satisfied, MABSplit scales as $O(mT\text{log}N)$. In many cases, this is much better than $O(m(N+T))$, e.g., for large datasets, because the dependence on $N$ is reduced from linear to logarithmic. More concretely, treating $T$ as a constant and ignoring the dependence on $m$, we reduce the complexity of the binned algorithm from $O(N)$, what we refer to as "linear", to $O(\text{log}N)$. We have added a clarification of this to the revision.
>
> We would also like to point out that in the worst case when all data points are evaluated, MABSplit reduces to the binning method; hence, its performance is never worse in sample complexity. In this case, we observe that MABSplit has slightly higher runtime than the binned algorithm due to the computational overhead of sampling the data in batches instead of all at once. This can happen, for example, on extremely small datasets or when all feature-value splits have identical impurity reductions.
>
> We agree that the original discussion of these points was unclear. We will clarify these points and make the distinctions between approaches more clear, as well as provide the pseudocode for all of these algorithms, in the final paper.
>
> [Note: we believe Reviewer AzdZ is using ``quadratic'' to refer to $O(mNT)$, which is $O(mN^2)$ when $T = O(N)$.
> MABSplit never scales quadratically in $N$, even when $T = O(N)$; rather, it scales as $O(mN)$ in this case.
> However, we emphasize that we are also interested in the case where $T$ is constant, in which MABSplit scales as $O(mT\text{log}N)$.]
>
> $\textbf{Reviewer AzdZ Comment 2:}$ The authors do not compare their approach against existing methods. More specifically, they do not compare it against the estimations used by Hoeffding's Trees [13,30] or the F-Forest [16] although these seem to be the only t[w]o similar methods available.
>
> $\textbf{Response to Reviewer AzdZ Comment 2:}$
>
> Thank you for your comments. We chose not to compare with Hoeffding Tree [13,30] because these methods focus on the online setting where datapoints arrive in real time and can only be evaluated once (i.e., performance is measured prequentially). In contrast, our work considers the offline setting. Therefore, our method is not directly comparable to Hoeffding Tree. However, we acknowledge it is an interesting direction to extend MABSplit to the online setting. We have added this point to the discussion section.
>
> Empirically, we attempted to train a Hoeffing Tree Classifier from the implementation in $\texttt{scikit-multiflow}$, a machine learning package for streaming algorithms, on the MNIST dataset. Even after extensive hyperparameter tuning and training time of over an hour, the Hoeffding Tree Classifier only results in test accuracy of only 59.64\%. The Random Forest Classifier from $\texttt{scikit-learn}$, with a single tree and no hyperparameter tuning, results in a test accuracy of 80.88\% within a minute.
>
> We chose not to compare to F-Forest [16] because it was not straightforward to implement this algorithm. Specifically, the paper did not provide pseudo-code or an open-source implementation. Furthermore, F-Forest's complexity is $O(N\text{log}N)$ in the number of samples $N$ (see Theorem 4.7 of [16]), the same as the standard RF algorithm. As such, we determined it is of low priority to compare our method to F-Forest.

---

> ### Author Response · Authors · 2022-08-02
> **Response to Reviewer AzdZ (1 of 3)**
>
> We thank Reviewer AzdZ for their detailed and insightful comments.
>
> $\textbf{Reviewer AzdZ Comment 1:}$ The authors refer to a "linear" runtime multiple times in the paper, but I don't know exactly what they mean by this... Please refine what you mean by "linear runtime" throughout the paper. It would probably best to just give some pseudo-code of the exact split induction (with or without binning) to which you compare against... You refer to "linear runtime" to the linear scan in binning and not the linear scan in the "standard approach". This is confusing because you basically always talk about binning, but then mention sorting in line 306 (in the conclusion)... To me, the runtime of the proposed method is somewhat unclear. The authors give a theorem with the expected runtime, however, there is no worst-case runtime? I think the worst-case runtime is at-least also quadratic which means that for certain use-cases the proposed method is worse than the standard approach implemented in many frameworks (see my question below)... Point out the worst-case runtimes and the expected runtime of your method and related work.
>
> $\textbf{Response to Reviewer AzdZ Comment 1:}$
> Reviewer AzdZ's understanding of baseline algorithms, described in Question 1, is all correct.
> We first provide a recapitulation of the baselines and their complexities here for clarity.
> We then describe the sample complexity of MABSplit in the context of baseline methods.
>
> We assume that we have a dataset with $N$ datapoints, $m$ features, and $T$ possible thresholds at which to split each feature. We discuss the classification setting for simplicity as per the Reviewer's suggestion, though the same arguments apply to regression.
>
> - The very naive approach would be to iterate over all $mT$ feature-value splits, and compute the probabilities $p_{L, k}$ and $p_{R,k}$ from all $N$ datapoints. This results in complexity $O(mTN)$, which is $O(mN^2)$ when $T = N$.
> - The "standard" approach sorts all $N$ datapoints in $O(N\text{log}N)$ time for each of the $m$ features, resulting in total computational cost $O(mN\text{log}N)$. Then the algorithm scans linearly from lowest value to highest value for each feature and update the parameters $p_{L, k}$ and $p_{R,k}$, via simple counting, to find the best impurity reduction for each of the $T$ potential splits. The complexity of this step is $O(mT + mN)$, where the "$+mN$" comes from the allocations of each data point to the left or right node during the scan (each data point is re-allocated only once per feature). Thus the total complexity of this approach is $O(mN\text{log}N + mT + mN) = O(mN\text{log}N + mT)$. This is $O(mN\text{log}N)$ when $T = N$.
> - The binning (a.k.a. histogramming) method, in particular when $T < N$, does not require the per-feature sort and avoids the $O(mN\text{log}N)$ computation. Instead, each of the $N$ points must be inserted into the correct bin (which can be done in $O(1)$ time for each datapoint if the bins are equally spaced) for each of the $m$ features, incurring total computational cost $O(mN)$. Then, the same linear scanning approach as in the ``standard'' algorithm is performed with complexity $O(mT + mN)$. The total complexity of this approach is $O(mN + mT + mN) = O(m(N+T))$. This is $O(mN)$ when $T = N$.
>
> In general, we do not assume $T = N$, i.e., that every feature value is a potential split point, unless otherwise specified. We will clarify in our paper that the "standard" approach refers to the $\underline{\textbf{un}}$binned approach which requires an $O(mN\text{log}N)$ sort, and that the binned approach is $O(m(N+T))$, which is $O(mN)$ when $ T = O(N)$, and include pseudocode for these algorithms in the final paper.

---

### Official Review · Reviewer_4wQQ · 2022-07-11

**Rating:** 5
**Confidence:** 4
**Soundness:** 2 fair
**Presentation:** 3 good
**Contribution:** 2 fair

**Summary:**

The algorithm considers the selection of decision stumps using bandit algorithms. The decision stumps are used in algorithm such as random forest, extremely randomized forests, or random patches. The proposed method allows for a faster exclusion of candidate stumps that are not looking promising after a smaller number of evaluations (using Gini impurity or entropy).  Theoretical results offer some guarantees on the computational complexity. Empirical evaluation on the MNIST dataset show that the proposed selection technique results in much faster tree algorithms.


**Questions:**

Framing the proposed method as a bandit algorithm seems a bit misleading, since all candidate stumps are evaluated. There is some connection with the arm elimination, but the differences in the setup should be perhaps better clarified.


**Limitations:**

As mentioned above, the proposed technique might be more difficult to implement/optimize effiently.


**Strengths And Weaknesses:**

Reducing the set of candidate stumps using statistical measures is natural. I am not aware of previous work published with similar effect, but I would not be surprised if it has been attempted before. The theoretical guarantees seem fairly straightforward.

The empirical results are fairly impressive. It would have been useful to replicate the results on different datasets.

The experiments use reimplemented variants of the baseline algorithms. I understand the authors' argument for this choice, nevertheless, it should be pointed out that the full evaluation of stumps makes the optimizations present in publicly available implementations much easier. These optimization techniques might not be possible (or more difficult) for the selection technique proposed in the paper. Therefore, it would have been useful to provide the walk clock time comparisons with the publicly available implementations as well.

---

> ### Author Response · Authors · 2022-08-02
> **Response to Reviewer 4wQQ**
>
> We thank Reviewer 4wQQ for their constructive suggestions.
>
> $\textbf{Reviewer 4wQQ Comment 1:}$ The empirical results are fairly impressive. It would have been useful to replicate the results on different datasets.
>
> $\textbf{Response to Reviewer 4wQQ Comment 1:}$
>
> Thank you for the comment. Following the reviewer suggestion, we have tested MABSplit's performance across three additional classification datasets and two additional regression datasets. The addenda to each of Tables 1, 2, 3, and 4 is presented in a comment to all reviewers and we reference them in the remainder of our response. We will add them to the final paper (though we have omitted it from the revision due to the original space limit of 8 pages).
>
> In the additional classification experiments, MABSplit demonstrates increasing benefit across the classification tasks as the dataset size increases, from approximately 10x speedups on the APS Failure Dataset ($N = 60,000$) to approximately 30-50x speedups on the Forest Cover Type Dataset ($N = 581,012$). MABSplit also demonstrates superiority over baselines in the additional regression experiments on the Beijing Air Quality Dataset and SGEMM GPU Kernel Performance Dataset.
>
> Across a single dataset, MABSplit demonstrates the predicted logarithmic scaling across various subset sizes in both classification and regression (see Appendix 2 in the original submission and revision).
>
> $\textbf{Reviewer 4wQQ Comment 2:}$ The experiments use reimplemented variants of the baseline algorithms. I understand the authors' argument for this choice, nevertheless, it should be pointed out that the full evaluation of stumps makes the optimizations present in publicly available implementations much easier. These optimization techniques might not be possible (or more difficult) for the selection technique proposed in the paper. Therefore, it would have been useful to provide the walk clock time comparisons with the publicly available implementations as well.
>
> $\textbf{Response to Reviewer 4wQQ Comment 2:}$
>
> Thank you for the comment. We appreciate the acknowledgement of our original arguments that our strategy of reimplementing the baseline algorithms may provide a more fair evaluation of the algorithmic improvement of our method, instead of competing on low-level implementation optimizations. We note that a similar approach focusing on sample complexity instead of wall-clock time has also been used in related work, such as NeurIPS '20 [40], AAAI '19 [Liu et al., "A Bandit Approach to Maximum Inner Product Search"], and  AISTATS '18 [4].
>
> We also agree that a wall-clock time comparison with publicly available implementations is important, but this requires us to fully optimize MABSplit based on techniques present in publicly available implementations such as those in $\texttt{scikit-learn}$. To do this, we need to integrate our algorithm into the complex $\texttt{scikit-learn}$ library and to perform a series of low-level optimizations such as cache optimizations and manual loop unrolling (we discuss these points in Lines 205-215 of the original submission). This may be out of the scope of the current work and we leave it as an interesting direction for future work. We have added a discussion of these potential limitations to Appendix 7 of the revision.
>
> $\textbf{Reviewer 4wQQ Comment 3:}$ Framing the proposed method as a bandit algorithm seems a bit misleading, since all candidate stumps are evaluated. There is some connection with the arm elimination, but the differences in the setup should be perhaps better clarified.
>
> $\textbf{Response to Reviewer 4wQQ Comment 3:}$
>
> We apologize for the confusion. We treat each candidate split (feature-threshold pair) as an arm. We provide a full description of the reformulation of the node-splitting problem as a multi-armed bandit problem in the table at https://imgur.com/a/4eNnrwE, reproduced above, and have added a clarification to the revision.
>
> We would also like to clarify that in all experiments of the main paper, we train trees of depth between $3$ and $5$ (i.e., not just a single node split for a decision stump) in each forest.
>
> Please let us know if you have any further questions and/or comments.

---

> > ### Comment · Reviewer_4wQQ · 2022-08-07
> > **Experiments**
> >
> > The additional experiments do improve the paper.

---

> ### Author Response · Authors · 2022-08-05
> **Table describing reduction of node-splitting problem to best-arm identification problem**
>
> | **Best-arm identification terminology** | **Corresponding term in node-splitting problem**                                                                    |
> |-----------------------------------------|---------------------------------------------------------------------------------------------------------------------|
> | Arm                                     | Feature-value threshold: $(f, t)$                                                                                   |
> | Arm parameter                           | Reduction in impurity for splitting the node at the given feature-value threshold: $\mu_{ft}$ described in Line 150 |
> | Pulling an arm                          | Drawing another datapoint from the dataset                                                                          |
> | Observing arm return                    | Calculating the induced change in impurity reduction $\hat{\mu}_{ft}$ from the new datapoint                        |

---

### Official Review · Reviewer_PuUT · 2022-07-11

**Rating:** 7
**Confidence:** 3
**Soundness:** 3 good
**Presentation:** 3 good
**Contribution:** 3 good

**Summary:**

In the paper "MABSplit: Faster Forest Training Using Multi-Armed Bandits", a splitting criterion for fitting decision trees is proposed based on multi-armed bandits as to improve training efficiency, i.e., the time to train an entire random forest. To this end tuples of features and thresholds are each considered an arm and pulled for an instance to build histograms. Thereby, a sublinear runtime complexity (wrt. to the number of instances) can be achieved instead of linear runtime.

**Questions:**

* How does MABSplit perform across various datasets?
* Why was only an artificial dataset used for regression?
* What are limitations of MABSplit?
* What are potential drawbacks of MABSplit?
* Why does the performance for ExtraTrees in Table 2 degrade that much?


**Limitations:**

Limitations are not discussed in the paper. Thus, it is not clear what the strengths and the weaknesses of the approach are.

**Strengths And Weaknesses:**

## Strengths
The problem addressed by the paper is an important one, especially with sustainability and the climate crisis in mind. Working towards more efficient algorithms reduces energy consumption but also makes the usage of RF even more responsive in the sense that models can be obtained more quickly.

To the best of my knowledge, while there are various works on different split criterions for decision trees, the approach of this paper is novel and presents an original idea.

Overall the paper is well structured and easy to read. However, I cannot induce the schema according to which the tables were colored and there is no explanation on that. Moreover there are still a few grammatical mistakes, e.g., l.22 "is then be aggregated", l.45 + l.241 "O(n) computation" -> computations, l. 112 the sentence is not complete, l.116 "a M" -> an M, l.224 "data point" -> data points. Furthermore sklearn's true name is scikit-learn.

## Weaknesses

While the general idea is quite interesting and the tackled problem is an important one, the paper falls short in terms of the scope of contribution. The theoretical contributions are relatively straightforward and the empirical study is quite limited. Although different variants of random forests are compared, only a single dataset (for a single split) for classification and a single artificial dataset for regression is used to benchmark and compare the proposed method with baselines. While this is already a good indicator that the method works indeed as intended, a more indepth evaluation of the proposed method is required to draw conclusions on a broader scale: How does MABSplit perform across various dataset sizes? For most tabular datasets MNIST already is comparably large, although for way smaller datasets there is not really a need for speeding up random forests. Anyways it would still be important to see how MABSplit compares for such smaller datasets. Then, the authors stated repeatedly that MNIST is only a small dataset. So, the question is how does it compare for datasets larger than MNIST? Does it scale as well as expected? For regression tasks the same arguments hold. Additionally, it is unclear why an artificial dataset was generated instead of using real-world data, e.g., from OpenML?

The result descriptions are quite descriptive without any more indepth explanations. The performance degradation in Table 2 was not addressed at all. How come that the performs degrades that much for ExtraTrees?

---

> ### Author Response · Authors · 2022-08-02
> **Response to Reviewer PuUT (2 of 2)**
>
> (Continued from above)
>
> $\textbf{Reviewer PuUT Comment 5:}$ What are limitations of MABSplit? What are potential drawbacks of MABSplit?
>
> $\textbf{Response to Reviewer PuUT Comment 5:}$
>
> Thank you for the question.  First, due to the overhead of the adaptive procedure, the traditional RF may be preferred for smaller datasets. Second, MABSplit relies on the assumption that the split gains are heterogeneous at each node across the candidate splits so that the adaptive procedure can effectively distinguish between good and bad splits, as detailed in Lines 237-245 of the original submission. However, MABSplit will not be worse than the non-adaptively version in sample complexity if such an assumption is violated. We have added a discussion of these limitations to Appendix 7 of the revision.
>
> $\textbf{Reviewer PuUT Comment 6:}$ Why does the performance for ExtraTrees in Table 2 degrade that much?
>
> $\textbf{Response to Reviewer PuUT Comment 6:}$ We apologize for the confusion. The performance degradation in Table 2 was due to a non-standard hyperparameter choice for ExtraTrees (the number of thresholds, $T$). After changing it to the recommendation in the original paper [17], we no longer observe the performance degradation. We provide an updated version of the ExtraTrees results in the table at https://imgur.com/a/RNyBBFL and have updated Table 2 in the revision.
>
> Please let us know if you have any further questions and/or comments.

---

> > ### Comment · Reviewer_PuUT · 2022-08-06
> > **Thanks for the Clarifications**
> >
> > Thank you very much for the clarifications. The additional experiments are very much appreciated.

---

> ### Author Response · Authors · 2022-08-02
> **Response to Reviewer PuUT (1 of 2)**
>
> We thank Reviewer PuUT for their insightful comments.
>
> $\textbf{Reviewer PuUT Comment 1:}$ I cannot induce the schema according to which the tables were colored and there is no explanation on that. Moreover there are still a few grammatical mistakes... Furthermore $\texttt{sklearn}$'s true name is $\texttt{scikit-learn}$.
>
> $\textbf{Response to Reviewer PuUT Comment 1:}$
>
> We thank the reviewer for pointing these out. Originally, we used green for the timing experiments (Tables 1 and 2), blue for the fixed-budget experiments (Tables 3 and 4), and red for the feature stability experiment (Table 5). We made all the tables white and fixed all of these points in the revision. We have also reviewed the manuscript for grammar and typographical errors.
>
> $\textbf{Reviewer PuUT Comment 2:}$ The theoretical contributions are relatively straightforward.
>
> $\textbf{Response to Reviewer PuUT Comment 2:}$
>
> For the theoretical contribution, we would like to clarify that our main contribution is to propose to use multi-armed bandits (MABs) to accelerate RF training, and not to propose a new MAB algorithm.
>
> We consider our reduction of the RF training procedure to a MAB problem non-trivial as it achieves state-of-art performance for RF. RFs are one of the most widely-used machine learning algorithms, and the research community is actively trying to optimize RF training [12, 16, 24, 30, 36, 43, 44]. Our improvements are demonstrated by both the theoretical sample complexity and the empirical results. We also note that using MABs to accelerate ML algorithms has witnessed success in recent machine learning conferences, such as NeurIPS '20 [40], AAAI '19 [Liu et al., "A Bandit Approach to Maximum Inner Product Search"], AISTATS '18 [4].
>
> $\textbf{Reviewer PuUT Comment 3:}$ The empirical study is quite limited. Although different variants of random forests are compared, only a single dataset (for a single split) for classification and a single artificial dataset for regression is used to benchmark and compare the proposed method with baselines. While this is already a good indicator that the method works indeed as intended, a more indepth evaluation of the proposed method is required to draw conclusions on a broader scale...[H]ow does it compare for datasets larger than MNIST?... For regression tasks the same arguments hold. Additionally, it is unclear why an artificial dataset was generated instead of using real-world data, e.g., from OpenML?... How does MABSplit perform across various dataset sizes? ... Does it scale as well as expected?
>
> $\textbf{Response to Reviewer PuUT Comment 3:}$
>
> Thank you for the comment. To address these comments, we have tested MABSplit's performance across three additional classification datasets and two additional regression datasets. The addenda to each of Tables 1, 2, 3, and 4 is presented in a comment to all reviewers and we reference them in the remainder of our response. We will add them to the final paper (though we have omitted it from the revision due to the original space limit of 8 pages).
>
> In the additional classification experiments, MABSplit demonstrates increasing benefit across the classification tasks as the dataset size increases, from approximately 10x speedups on the APS Failure Dataset ($N = 60,000$) to approximately 30-50x speedups on the Forest Cover Type Dataset ($N = 581,012$). As expected, MABSplit's benefit increases as the dataset size increases. MABSplit also demonstrates superiority over baselines in the additional regression experiments on the Beijing Air Quality Dataset and SGEMM GPU Kernel Performance Dataset.
> The additional experiments demonstrate MABSplit's benefits across a variety of real-world datasets in both classification and regression.
>
> Across a single dataset, our theory predicts that MABSplit scales logarithmically with dataset size. We provide empirical validation of the predicted logarithmic scaling in a classification task in Appendix 2 of the original submission. We have also added similar results from a regression task in Appendix 2 of the revision. We provide both plots in Figure 1 of the supplemental new figures.
>
> We would also like to clarify that in all experiments of the main paper, we train trees of depth between $3$ and $5$ (i.e., not just a single node split for a decision stump) in each forest.
> We have made this more explicit in the revision.
>
> $\textbf{Reviewer PuUT Comment 4:}$ It would still be important to see how MABSplit compares for such smaller datasets.
>
> $\textbf{Response to Reviewer PuUT Comment 4:}$
>
> We have added experiments on small subsets of MNIST in Figure 2 of our new supplemental figure submission. We demonstrate that RF+MABSplit outperforms the standard RF algorithm, in both sample complexity and wall-clock time, when the dataset size exceeds approximately $1100$. However, we also note that the main use case for MABSplit is when the data size is large and it is computationally challenging to run the standard RF models.

---

### Official Review · Reviewer_wtDT · 2022-07-11

**Rating:** 6
**Confidence:** 3
**Soundness:** 2 fair
**Presentation:** 3 good
**Contribution:** 3 good

**Summary:**

In this paper, they proposed a MAB application for tree-based learning methods. Finding the best split takes O(n log(n)) time for a fixed feature, where n is the number of training instances. Their proposed algorithm, MABSplit, estimates the best split by drawing n' < n independent examples. MABSplit works as follows;
1. Define the feasible set as the set of pairs of features and thresholds.
2. While the size of the feasible set is greater than 1:
    a. Draw a batch sample from training instances.
    b. Shrink the feasible set.
They evaluated the computational complexity and verified the effectiveness of their algorithm by experiment.

**Questions:**

- Correctness of algorithm 1 (please see the details at limitations part).

**Limitations:**

I have some concerns about the correctness of the paper. Estimating entropy or gini index via sampling can be only done approximately. In particular, the entropy changes significantly at both ends (0 and 1), and the gradient is infinitely large there. So, it would need a careful analysis to employ Hoeffding bounds and others (e.g., case analysis). Furthermore, to ensure the correctness of algorithm 1, it would need an assumption about the gap between gains of the best one and others.

**Strengths And Weaknesses:**

Strength:
- a new application of bandit techniques to decision tree learning


Weakness:
- sone more explanation is necessary for the correctness of the algorithm

---

> ### Author Response · Authors · 2022-08-02
> **Response to Reviewer wtDT**
>
> We thank Reviewer wtDT for their time and insightful questions on theoretical aspects of the algorithm. We provide a point-by-point response below.
>
> $\textbf{Reviewer wtDT Comment 1:}$ Estimating entropy or gini index via sampling can be only done approximately. In particular, the entropy changes significantly at both ends (0 and 1), and the gradient is infinitely large there. So, it would need a careful analysis to employ Hoeffding bounds and others (e.g., case analysis).
>
> $\textbf{Response to Reviewer wtDT Comment 1:}$
> We thank the reviewer for pointing this out.
> The use of the delta method indeed requires a careful consideration of the gradient of the different impurity measures we consider. Let $n$ be the number of samples. Our key assumption for Theorem 1 is that the $(1-\delta)$ confidence interval scales as $O(\sqrt{\frac{\text{log}\frac{1}{\delta}}{n}})$ (Lines 222-224 of the original submission). For Gini impurity, the gradient of the impurity with respect to each $p_k$ is finite and bounded, which allows us to apply the delta method for finite confidence intervals and achieve this scaling (see Appendix 3 of the original submission and revision).
>
> For entropy, however, as pointed out by the reviewer, the gradients may be unbounded when some of the $p_k$'s are close to zero. In such cases, the delta method used in the paper will give wide confidence intervals. As a result, MABSplit may not be able to efficiently eliminate unpromising candidate splits, and may reduce to its non-adaptive counterpart. This issue should not have a major impact because it usually occurs at the later stages of the decision tree construction process where the number of samples is small and some of the $p_k$'s may become zero. In these cases, since the number of samples is small, so falling back to the non-adaptive counterpart will not substantially hurt the performance. In addition, one can use alternative methods to construct the confidence intervals for entropy that achieve the desired scaling, e.g., [*] and [**]. We have added a discussion of these points to Appendix 7 of the revision.
>
> [*]: G. P. Basharin, "On a Statistical Estimate for the Entropy of a Sequence of Independent Random Variables"
>
> [**]: L. Paninski, "Estimation of Entropy and Mutual Information".
>
> $\textbf{Reviewer wtDT Comment 2:}$ Furthermore, to ensure the correctness of algorithm 1, it would need an assumption about the gap between gains of the best one and others.
>
> $\textbf{Response to Reviewer wtDT Comment 2:}$
>
> Reviewer wtDT is correct that an assumption must be made about the gap between gains of the best split and the others. These assumptions are common in the literature (e.g., [4, 6, 13, 30, 40]) and are described in Lines 236-244 of the original submission. More specifically, we avoid the adversarial case, in which many arms have arbitrarily close gains to that of the best arm, by making assumptions about the distribution of the arm returns. For example, when the arm rewards $\mu_{ft}$'s follow a sub-Gaussian distribution across all arms $(f, t)$, MABSplit will have an overall complexity of $O(mT \log(n^2mT))$.
>
> Our empirical results also demonstrate that such conditions hold in practice; in Appendix 2 of the original submission, we provide a plot demonstrating that this gap condition is satisfied and that MABSplit demonstrates logarithmic scaling in a classification task. In the corresponding appendix of the revision, we also demonstrate these results in a regression task. We provide both plots in Figure 1 of the new supplementary figures document.
>
> Please let us know if you have any further questions and/or comments.

---

> ### Author Response · Authors · 2022-08-08
> **Any additional questions?**
>
> We would like to thank Reviewer wtDT once again for their time and feedback on our paper. Please let us know if there are additional questions that we may address during the author-reviewer discussion period. We are happy to engage in further discussion as the discussion period allows.

---

### Author Response · Authors · 2022-08-02
**Additional Experimental Results for all Reviewers**

Here, we provide a comment to all reviewers with additional experimental results. We provide links to LaTeX-rendered figures and tables as we are unable to render them correctly in OpenReview.


$\textbf{Supplementary Figures:}$
https://imgur.com/a/dmVw1ZG contains two additional figures.

Figure 1 demonstrates the scaling of MABSplit across data subset sizes for in both a classification and a regression task. MABSplit demonstrates the expected logarithmic scaling in dataset size.

Figure 2 demonstrates the performance of MABSplit versus the brute-force node-splitting algorithm for small dataset sizes. For dataset sizes larger than approximately $1100$ datapoints, MABSplit outperforms the brute-force algorithm in wall-clock training time and sample complexity.

$\textbf{Supplementary Tables:}$
https://imgur.com/a/xuloNpF provides addenda to Tables 1, 2, 3, and 4 of our original paper.

Collectively, the tables extend our original submission's experimental results to three new classification datasets and two new regression datasets.

Appendix Table 1 demonstrates that MABSplit reaches the same performance as the brute-force solution, but usually much faster, across three classification datasets.

Appendix Table 2 demonstrates that MABSplit reaches the same performance as the brute-force solution, but usually faster, across two regression datasets.

Appendix Table 3 demonstrates that, under a fixed computational budget, models trained with MABSplit generally outperform those trained with the brute-force algorithm in test accuracy in several classification tasks.

Appendix Table 4 demonstrates that, under a fixed computational budget, models trained with MABSplit generally outperform those trained with the brute-force algorithm in test MSE in several regression tasks.

EDIT: We have also run wall-clock time comparisons of our implementation with $\texttt{scikit-learn}$; our implementation is about 4x faster on the MNIST digit classification dataset. Please see the table at https://imgur.com/a/NqSdoo3 for a full discussion.

---

### Meta-Review · Area_Chair_1Rps · 2022-08-28

**Recommendation:** Accept
**Confidence:** Certain

**Metareview:**

The reviewers agree that from a decision tree induction point of view, this paper provides a solid methodological approach in contrast to prior heuristics-based approaches. They found the author feedback satisfying, specifically, the additional experiments showcase that the proposed method generalizes most likely beyond development datasets.

We ask the authors that in the final version, please make a pass over the paper to incorporate the author-reviewer discussions.

**Award:**

No

---

### Decision · Program_Chairs · 2022-09-14

Accept